# Ecology, more than antibiotics consumption, is the major predictor for the global distribution of aminoglycoside-modifying enzymes

**Léa Pradier\*, Stéphanie Bedhomme**

CEFE, CNRS, Univ Montpellier, EPHE, IRD, Montpellier, France

**Abstract** Antibiotic consumption and its abuses have been historically and repeatedly pointed out as the major driver of antibiotic resistance emergence and propagation. However, several examples show that resistance may persist despite substantial reductions in antibiotic use, and that other factors are at stake. Here, we study the temporal, spatial, and ecological distribution patterns of aminoglycoside resistance, by screening more than 160,000 publicly available genomes for 27 clusters of genes encoding aminoglycoside-modifying enzymes (AME genes). We find that AME genes display a very ubiquitous pattern: about 25% of sequenced bacteria carry AME genes. These bacteria were sequenced from all the continents (except Antarctica) and terrestrial biomes, and belong to a wide number of phyla. By focusing on European countries between 1997 and 2018, we show that aminoglycoside consumption has little impact on the prevalence of AME-gene-carrying bacteria, whereas most variation in prevalence is observed among biomes. We further analyze the resemblance of resistome compositions across biomes: soil, wildlife, and human samples appear to be central to understand the exchanges of AME genes between different ecological contexts. Together, these results support the idea that interventional strategies based on reducing antibiotic use should be complemented by a stronger control of exchanges, especially between ecosystems.

## Editor's evaluation

This work presents important findings on the role of ecological niches in shaping the distribution of antibiotic resistance genes in the environment and human populations. The evidence supporting the conclusion is compelling, presenting a rigorous analysis of a large dataset with many possible confounding variables. The work should be of broad interest to microbiologists, public health workers, and policymakers.

**\*For correspondence:**
leaemiliepradier@gmail.com

**Competing interest:** The authors declare that no competing interests exist.

## Introduction

Antibiotic resistance (AR) is a persistent global public health problem that has increased over the last decades, with resistances spreading faster and faster after antibiotic introduction in clinical use (*Witzany et al., 2020*). Although first concerns about infections by antibiotic resistant bacteria (AR bacteria) were formulated in the 1940s, the discovery and development of new antibiotics allowed for treatment substitutions (*Podolsky, 2018*) during the first decades of antibiotic use. The discovery that AR was frequently acquired by horizontal gene transfer (HGT, *Watanabe, 1963*) and the subsequent emergence of multi-resistant strains led the international health agencies to start raising the issue, at the end of the 1970s, that antibiotic resistance propagation was threatening to end the antibiotic golden area and jeopardize the huge progress made in the treatment of bacterial infectious

diseases. In parallel, the discovery and design of new antibiotics had become more and more difficult (*Livermore et al., 2011*). This trend persisted despite the bulk of information provided by recent advances in genomics (*Ribeiro da Cunha et al., 2019*), thus decreasing the hope for potential treatment substitutions. A recent review (*Antimicrobial Resistance Collaborators, 2022*) evaluated that in 2019, around 1.25 million deaths were directly attributable to bacterial antimicrobial resistance.

Antibiotic consumption and its abuses have been historically and repeatedly pointed out as the major cause of AR genes propagation (*O'Neill, 2014*; *Podolsky, 2018*; *Schrijver et al., 2018*): the frequency of AR increases in bacteria communities under the selective pressure of antibiotics. To fight this threat, most health agencies are thus focusing their policies on sanitation and mostly on a more reasonable use of antibiotics (*World Health Organization, 2015*). This perception of the factors driving AR spread and its associated policies still prevail nowadays: for example, a meta-analysis over 243 studies found a positive correlation between antibiotic consumption and presence of AR (*Bell et al., 2014*). However, even if antibiotic consumption decreases in most developed countries, AR does not always follow the same path: for example, a drastic reduction in sulfonamide consumption in the United Kingdom during the 1990s was not followed by a decrease in prevalence of sulfonamide-resistant *Escherichia coli* (*Enne et al., 2004*). Antibiotic consumption therefore does not seem to be the only factor maintaining AR in pathogenic bacteria communities. Indeed, although the impact of human activities on the circulation and spread of AR genes is now well documented through the accumulation of specific examples, integrative studies identifying large-scale trends are lacking and this absence of general view has been pinpointed as a gap in our knowledge of drivers of antimicrobial resistance (*Holmes et al., 2016*).

Another strongly overlooked factor is ecology, and a growing number of studies has called for a more comprehensive analysis of AR outside of farms and hospitals (see e.g *Bengtsson-Palme et al., 2018*). Though data in natural ecosystems remain scarce, AR genes have probably always been natural members of the gene pools of environmental microbial communities (*D'Costa et al., 2011*). Natural ecosystems may contribute to the spread of AR, both as sources and as vectors of propagation (*Bengtsson-Palme et al., 2018*; *Berendonk et al., 2015*; *Marti et al., 2014*). This realization, combined with an increasing access to genomic data, led to bioinformatic studies where the goal was to extend our understanding of AR in ecological contexts that are often overlooked: for example β-lactam resistance in dairy industry (*Pitta et al., 2016*), in slaughterhouses (*Lavilla Lerma et al., 2014*), in wastewater treatment plants (*Karkman et al., 2016*), or in natural fresh water (*Czekalski et al., 2015*). However, even if these studies complement our knowledge on the presence, the frequency, the nature, and the circulation of AR in poorly documented environments (see e.g. *Cuadrat et al., 2020*; *Zhang et al., 2020*), descriptions of global distribution patterns and analyses of factors underlying them remain scarce.

In this study, we investigate the relative importance of environmental and genomic factors in shaping the routes of antibiotic spread on a large scale, focusing on aminoglycoside resistance. Aminoglycosides are antibiotics that bind to the 30 S subunit of prokaryotic ribosomes and thus inhibit protein synthesis (*Mingeot-Leclercq et al., 1999*). This antibiotic class was first introduced in 1944 with the successful use of streptomycin against tuberculosis, and the first antibiotic able to fight Gram-negative bacteria. Several years later, other drugs produced by *Streptomyces spp.* were discovered (kanamycin, spectinomycin, tobramycin, neomycin, apramycin) and introduced in clinics. They were followed in the 1970s by a series of new isolates or derivatives synthesized compounds (amikacin, netilmicin, isepamicin, dibekacin, arbekacin, see *van Hoek et al., 2011*). However, the emergence of resistant strains during the following years, combined with the requirement of administration by injection and possible nephro- and ototoxicity has reduced the use of aminoglycosides in therapies (*Murray and Murray, 1991*). Nowadays, these drugs are only used in humans as a second-line or last-resort treatment for Gram-negative bacteria (*Garneau-Tsodikova and Labby, 2016*). However, they remain frequently used in agriculture and veterinary medicine (*European Medicines Agency, 2019*). As an example, in 2015, aminoglycosides still represented 3.5% of the total sales of antimicrobials for farm animals, and they are still important drugs for some pathologies, for example post-weaning diarrhoea in pigs (*van Duijkeren et al., 2019*). Thus, aminoglycoside resistance is still highly prevalent among farm animals: in Denmark, 85% of *Salmonella spp. serovar Kentucky* isolated in turkeys are not susceptible to aminoglycosides, and in Italy, up to 67% of *E. coli* samples are resistant to at least one aminoglycoside drug (*van Duijkeren et al., 2019*). Thus, although these drugs are no longer widely prescribed for patients,

aminoglycoside resistance is still a major threat, at least toward food production and the treatment of patients infected by multi-resistant bacteria. Moreover, since most aminoglycosides originate from soil-dwelling bacteria, there are at least three ecological contexts (hospitals, farms, and soil) in which aminoglycoside resistance can evolve and spread to cause public health issues.

The three main mechanisms of aminoglycoside resistance are (*Garneau-Tsodikova and Labby, 2016*): (i) decrease in drug uptake (through modifications of membrane permeability or of the peri-membrane ion gradient) and/or increase in drug efflux (through efflux pump activation); (ii) drug inactivating enzymes; and (iii) modification of the drug target, for example by point mutations in the genes coding for ribosomal small subunit (*Finken et al., 1993*; *Sander et al., 1995*; *Toivonen et al., 1999*). Aminoglycoside-modifying enzymes (AMEs) are a class of inactivating enzymes that catalyze the transfer of chemical groups on specific residues of the aminoglycoside molecules, leading to a modified drug which poorly binds to its target (*Jana and Deb, 2006*). AMEs represent the most common mechanism of aminoglycoside resistance in clinical isolates and are well characterized biochemically (*Ramirez and Tolmasky, 2010*). The classical nomenclature of AMEs is based on the group they transfer (i.e. acetyltransferases, AACs; nucleotidyltransferases, ANTs; and phosphotransferases, APHs), on the residue modified, and on the resistance profile they confer (*Ramirez and Tolmasky, 2010*). However, AMEs emerged several times during evolution (see e.g. *Salipante and Hall, 2003* for AAC(6') enzymes), so biochemical nomenclatures do not reflect the evolutionary history of any class of AMEs. Finally, many AME genes are carried by mobile genetic elements, which give them the potential to be transmitted both vertically and horizontally (*Davies, 1983*). They represent today a major threat for the treatment of multidrug-resistant bacteria, notably *Mycobacterium tuberculosis* (*Labby and Garneau-Tsodikova, 2013*).

Through a computational approach, more than 160,000 publicly available genomes were screened to identify the presence of AME-encoding genes (AME genes) across the phylogeny of Eubacteria. The present study intended (i) to describe the genomic, geographical and ecological distribution of AME genes; and based on these data, (ii) to quantify the relative contribution of several key factors (geography, ecology, genomic context, human activities) potentially driving the spread of AME genes.

## Results

### Aminoglycoside resistance is widespread across geography, ecology, and phylogeny

160,987 publicly available Eubacteria genomes were screened for the presence of AME genes. The list of genomes and their metadata are listed in *Source data 1*. Published sequences of AMEs and genes coding for AMEs had been previously grouped in 27 clusters of homologous genes (CHGs), each containing sequences of genes and proteins very likely to share a common ancestor (see Materials and methods).

A total of 46,053 AME genes were detected in 38,523 genomes (i.e. about one quarter of the genomes screened). Their distribution in 27 CHGs is listed in *Table 1*, all the gene coordinates are listed in *Source data 2*, and their distribution over time, phyla, geography and ecology in *Source data 3*. Our dataset included 54 phyla across the phylogeny of Eubacteria but 89.6% of genomes belong to the three most represented phyla: Proteobacteria, Firmicutes, and Actinobacteriota. In the same way, though we analyzed genomes from 13,879 species, only 10 species made up 43.6% of the dataset. This imbalance was even stronger regarding the repartition of resistance genes: we found AME genes in 23 phyla, 97.2% of them were detected in the three most represented phyla. Moreover, whereas 23.9% of all the genomes in our dataset carried at least one AME gene, this proportion decreases to 16.3% across the least sampled species making up 50% of the dataset, and even to 10.9% across the species making up 10% of the dataset. The frequency of the resistance-carrying genomes presented very contrasting patterns across CHGs and phyla (*Figure 1*). For each CHG, the phylogenetic diversity of the species in which it was detected was evaluated by calculating Faith's distances (*Faith, 1992*). This reveals that the number of species carrying these genes ranges from 2 to 468 (respectively for ANTc with Faith's distance $d_{Faith} = 1.4$ and ANTa with $d_{Faith} = 78.4$), but also that for CHGs present in numerous species, these species can belong to a small number of phyla (e.g. AACc, $d_{Faith} = 24.5$ for 406 species) or be largely spread across the bacteria phylogenetic tree (e.g. AACf1, $d_{Faith} = 66.7$ for

**Table 1.** List of CHGs by biochemical function.

| Cluster | Biochemical function | Number of genes identified |
|---|---|---|
| AACa | N-acetyltransferases | 619 |
| AACb | N-acetyltransferases | 711 |
| AACc | N-acetyltransferases | 4636 |
| AACd | N-acetyltransferases | 1044 |
| AACe1 | N-acetyltransferases | 5230 |
| AACe2 | N-acetyltransferases | 13 |
| AACf1 | N-acetyltransferases | 11227 |
| AACf3 | N-acetyltransferases | 10 |
| AACg | N-acetyltransferases | 4016 |
| AACh | N-acetyltransferases | 3124 |
| AACi | N-acetyltransferases | 871 |
| AACj | N-acetyltransferases | 2246 |
| AACm | N-acetyltransferases | 50 |
| ANTa | nucleotidyltransferases | 4508 |
| ANTb | nucleotidyltransferases | 2990 |
| ANTc | nucleotidyltransferases | 21 |
| ANTd | nucleotidyltransferases | 982 |
| ANTe | nucleotidyltransferases | 59 |
| APHa | phosphotransferases | 675 |
| APHb | phosphotransferases | 351 |
| APHc | phosphotransferases | 49 |
| APHd1 | phosphotransferases | 518 |
| APHd2 | phosphotransferases | 67 |
| APHe | phosphotransferases | 20 |
| APHf | phosphotransferases | 1970 |
| APHg | phosphotransferases | 23 |
| APHh | phosphotransferases | 23 |

418 species). Regarding the phyla, some present a high diversity of CHGs (e.g. Proteobacteria, Actinobacteria, Firmicutes I) whereas others only contain a very small number of different CHGs.

The location and biome metadata could be recovered for 45,574 genomes and from these data, it was established that AME genes were present in samples coming from all the continents, with the exception of Antarctica. In most regions, the prevalence of AME-gene-carrying bacteria (AME bacteria) was between 20% and 40%, but it ranged from 10% in Japan, Eastern Europe, and Eastern Africa to 50% in Indonesia, Mexico, and Turkey (*Figure 2A*). A quite high spatial heterogeneity in terms of the proportion of each CHG among the AME bacteria was also revealed (*Figure 2B*): AACf1 is over-represented in the Southern hemisphere (Africa, South-East Asia, Oceania, Brazil), APHf is over-represented in Canada and Mexico and at very low proportion elsewhere, whereas Western Europe, the United States, and Japan have a rather balanced representation of all CHGs. Moreover, this heterogeneity does not appear to be structured by geographical distance itself: there are large differences when making comparisons e.g. between Canadian and the United States' resistomes, or between Western European and Central European resistomes. AME bacteria were identified in samples coming from all the biomes investigated. The vast majority of them come from clinical

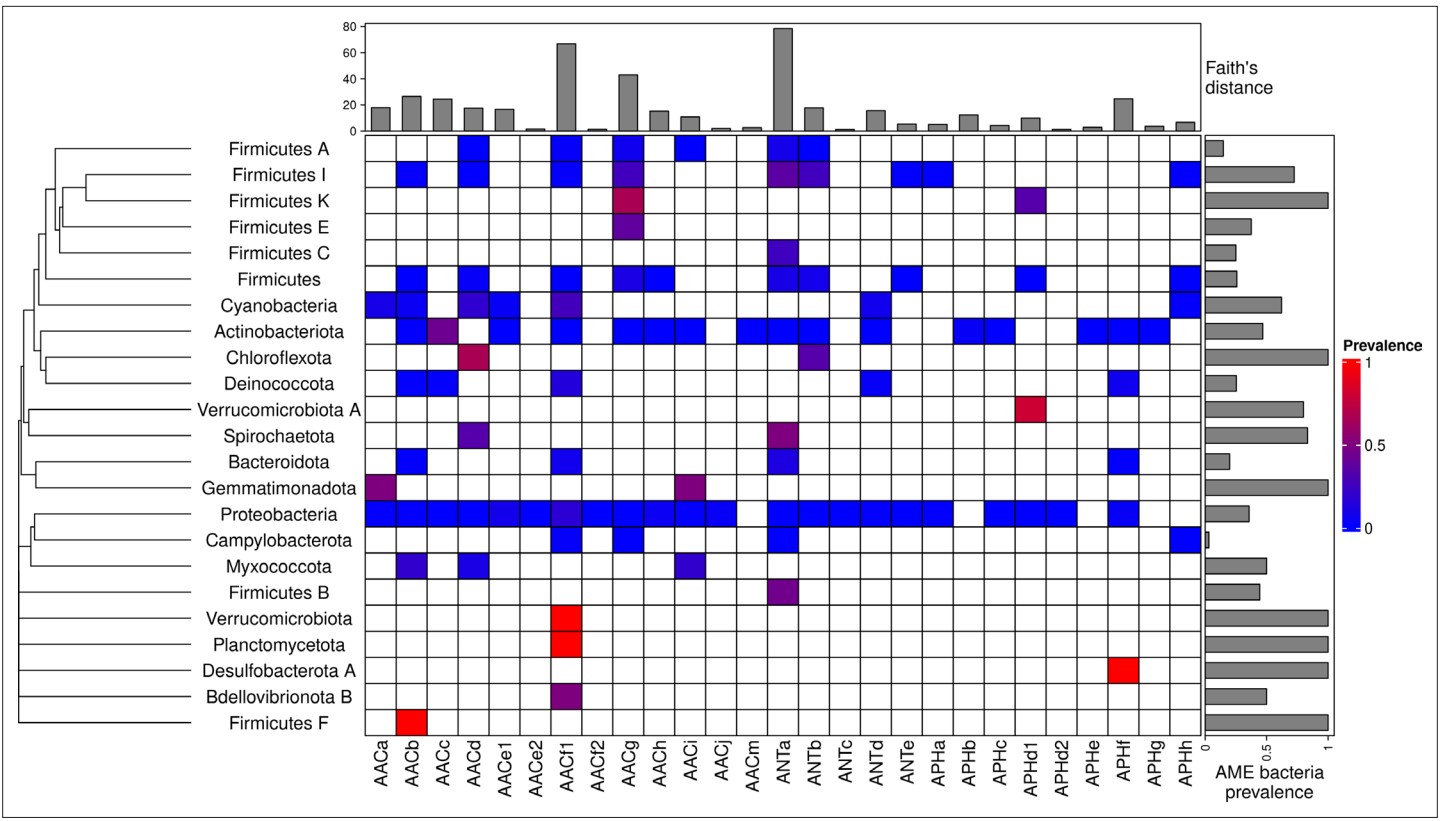

**Figure 1.** Prevalence of AME-gene-carrying bacteria across the phylogeny of Eubacteria. The phylogenetic tree corresponds to an aggregation of *bac120* phylogeny (**Parks et al., 2018**) to the phylum level. In the heatmap, blank boxes correspond to the observed absence of a CHG in a phylum. For the other colors, blue to red boxes stand for CHG frequencies from near-zero to one. Gray bars in the top part correspond to the Faith's distance (the sum of the lengths of all the branches on the bac120 tree) for the species in which each CHG was found. Gray bars on the right correspond to the prevalence of AME bacteria for each phylum, i.e. the proportion of genomes in which at least one AME gene was found.

samples (55.3%), human samples (22.1%), and farm samples (12.3%). Despite this large bias in ecological distribution, the prevalence of AME bacteria per biome varies in a relatively narrow range from 9% of bacteria sampled in domestic animals, to nearly 40% of bacteria sampled in humans (**Figure 3**). The ecological spread of CHGs is strongly correlated to their phylogenetic spread (Pearson correlation between the number of ecosystems and $d_{Faith}$, $\rho = 0.759$, p=6.8.10$^{-6}$). Thus, the CHGs with the largest $d_{Faith}$ are ecologically ubiquitous (AACc, AACe1, AACf1, AACg, ANTa and APHf), while 8 other CHGs with limited phylogenetic diversity were found in at most three biomes (AACe2, AACf3, ANTc, APHc, APHd2, AACm, ANTe, and APHg). Interestingly, clinical samples were found to carry resistance genes of all CHG except two and these two CHGs were specific to agrosystems only (APHg) and agrosystems and farms (AACf3). Moreover, there are significant negative correlations between the date of first sampling and the phylogenetic and ecological spreads (respectively $\rho = 0.712$, p=4.4.10$^{-5}$ and $\rho = 0.718$, p=3.6.10$^{-5}$): all ecologically ubiquitous CHGs were sampled before 1955, while the CHGs that were never sampled before 2000 could be found in at most four biomes.

Our dataset includes bacteria sampled between 1885 and 2019. However, the vast majority of them were sampled recently: 96.3% of genomes were sampled after 1990, and 58.9% after 2010. In this dataset, the first occurrence of an AME bacterium dates from 1905 (**Figure 4A**), i.e. before the human discovery of streptomycin in 1943. Moreover, the prevalence of AME bacteria remains constant (about 30%) between 1990 and 2019 (**Figure 4B**), a period during which no new aminoglycoside drug was released for commercial use. However, the frequencies of individual CHGs varied much more over time in this same time frame (**Figure 4C**), with some CHGs having a regularly increasing (e.g. AACe1, AACh) or decreasing (e.g. AACf1) frequency over the whole period or showing one-time frequency peak (e.g. ANTb in 2003 and 2004). The worldwide dynamics shows more a coexistence across time of a diversity of CHG than sequential substitution of one CHG by another. However, though we also

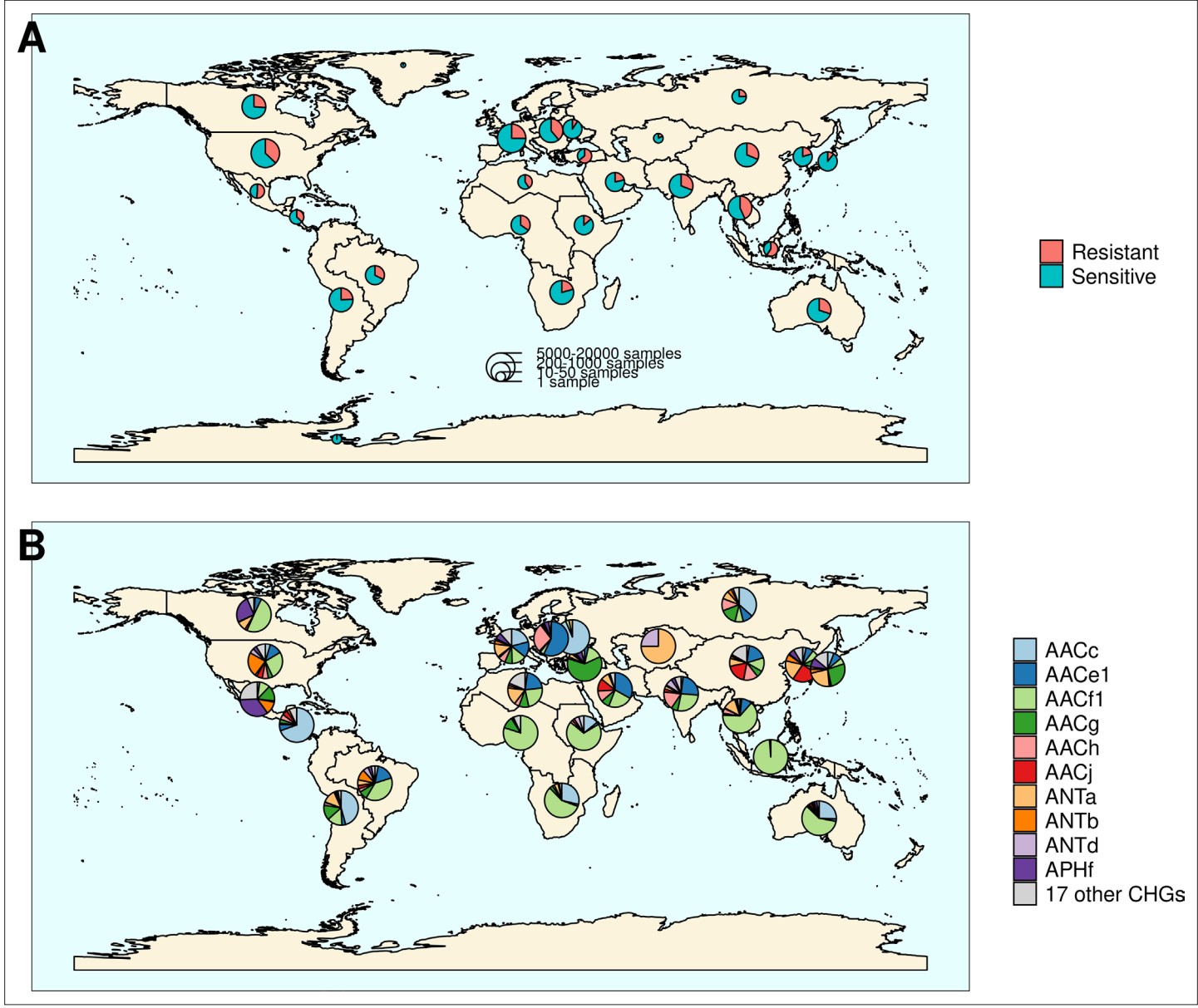

**Figure 2.** Distribution of aminoglycoside-resistant bacteria and AME-encoding genes over the world. (**A**) Distribution of sampled aminoglycoside-resistant bacteria in the world. The frequencies of resistant and sensitive bacteria are displayed in red and blue, and the size of pies represents the number of genomes sampled in a given region. (**B**) Distribution of sampled AME genes in the world. Pie size is irrelevant to the number of samples.

observe coexistence across time at continental levels (*Figure 4—figure supplement 1*), local time trends suggest that some CHGs might be progressively replaced: for example the frequency of AACc and AACe1 has increased a lot at the expense of AACf1 in Europe, Southeastern Asia and Oceania since the 1990s. This moreover shows that, despite geographical differences between continental resistomes (see *Figure 2B*), these differences are not consistent over time.

## The European distribution of aminoglycoside resistance is driven by ecology and human exchanges

We investigated the potential role of different factors on the distribution of AME bacteria in Europe: ecology (structuration in biomes), human exchanges (potential AME bacteria importations through immigration and merchandise imports), and aminoglycoside consumption. This analysis was performed on the timeframe 1997–2018 for which antibiotic consumption data by country and by year

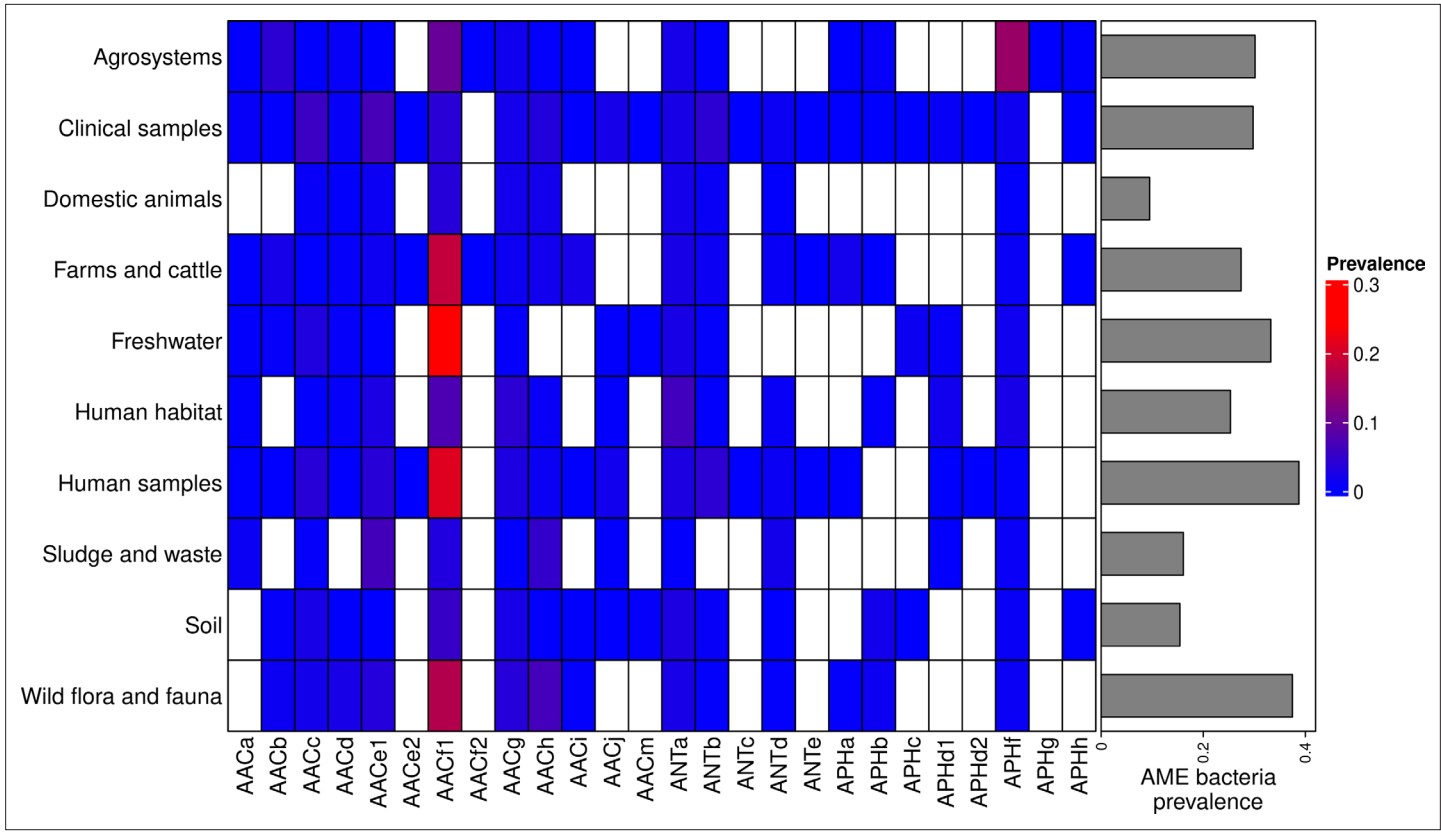

**Figure 3.** Prevalence of AME-gene-carrying bacteria across land biomes. In the heatmap, blank boxes correspond to the observed absence of a CHG. For the other colors, blue to red boxes stand for CHG frequencies from near-zero to 0.3. Gray bars on the right correspond to the prevalence of AME bacteria for each biome, i.e. the prevalence of genomes in which at least one AME gene was found. APHe is excluded from the heatmap because it was sampled in seawater only.

was available. It is important to note here that during this period, aminoglycoside consumption was quite constant over time, but varied strongly between countries. We included in the dataset 16 CHGs for which occurrences were detected in at least 30 genomes sampled in Europe during this timeframe. Most CHGs had very distinct distributions over time and space, so they were analyzed separately.

## Analysis of the whole dataset

Within Europe, a Matérn spatial autocorrelation structure was kept in all the models: CHGs tend to cluster between close countries and the autocorrelation structure allowed to control for this. A time autoregressive structure was also kept in all the models, which showed positive autocorrelation for half of CHGs (the others show negative time autocorrelation). On average, the inputs of the dataset reduced deviance by $R^2_{adj}$=32.2% (ranging from 10.1% to 67.6%). Ecology, human exchanges, and antibiotics consumption are kept as explanatory variables for respectively 16 CHGs, 13 CHGs, and 10 CHGs out of 16. The importance of each explanatory variable varies across CHGs, but ecology overall appears to be the most important one: it increased adjusted $R^2_{adj}$ by 24.4% on average (4.6% on average for human exchanges, and 2.9% on average for antibiotics consumption). Interactions between ecology and human exchanges increased $R^2_{adj}$ by 6.9% (for 6 CHGs), and interactions between ecology and antibiotics consumption increased $R^2_{adj}$ by 2.6% (for 6 CHGs) (*Figure 5*). The effects of all variables and their interactions for individual CHGs are given in *Supplementary file 1*.

The effect of AG consumption is not unidirectional across all the CHGs: when kept in the selected model, its effect is significantly positive for only three CHGs (AACa, AACc, and AACe1), but negative for three others (AACf1, APHd1, and APHf) and nonsignificant for the others. Human exchanges represented in the analysis by our proxy for potential AME bacteria influx due to trade, likewise has a positive significant effect on the AME bacteria prevalence for few CHGs (AACi, AACj, ANTb, and

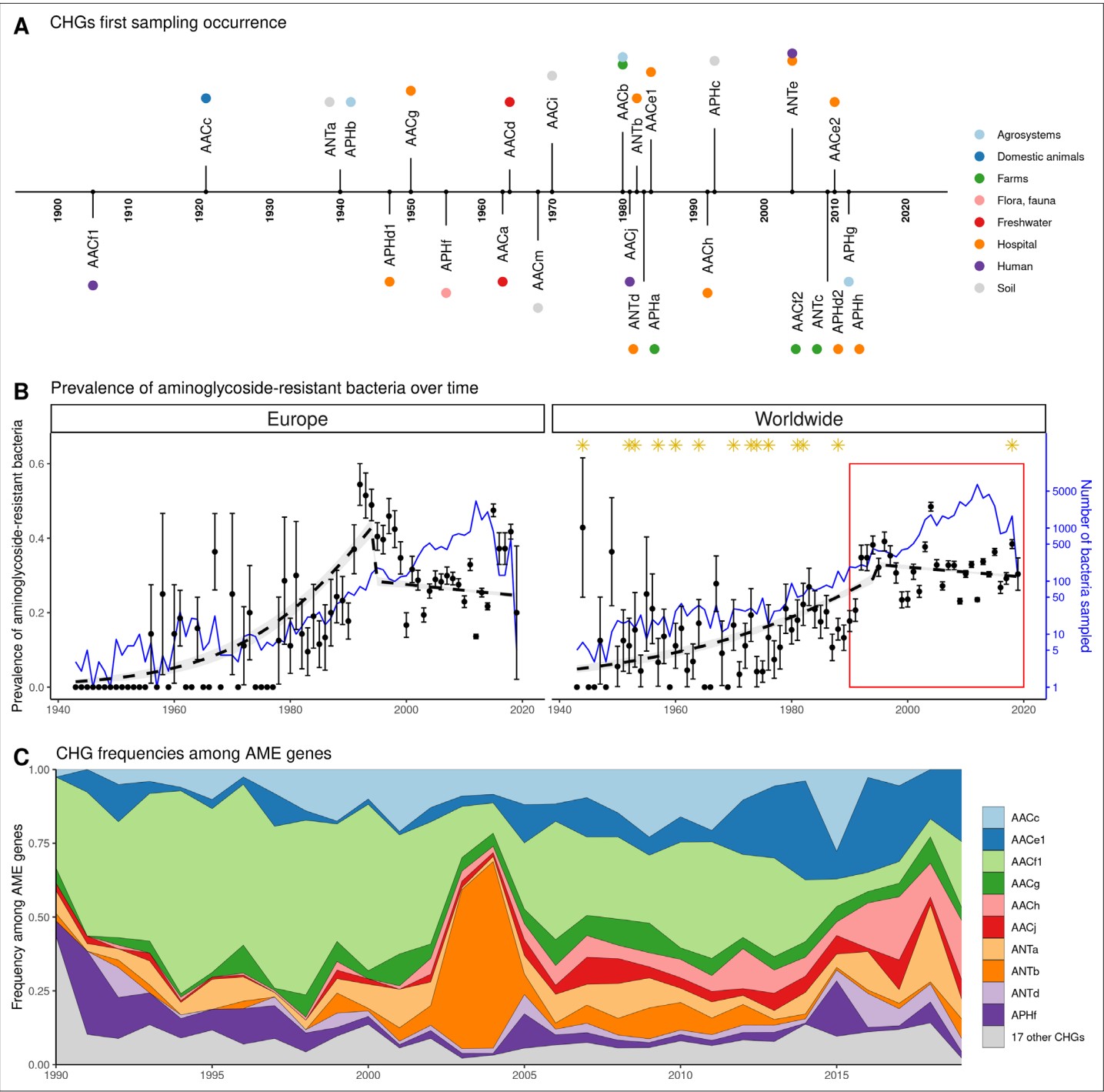

**Figure 4.** Global time trends for aminoglycoside resistance. (**A**) First sampling occurrence for each CHG in the analyzed dataset. First sampling occurrence of APHe is unknown. (**B**) Evolution of the worldwide and European prevalences of AME bacteria between 1943 and 2019. The dots and error bars represent the measured prevalence (± standard error) of AME bacteria each year. The dotted line represents a binomial regression of the prevalence of AME bacteria over time, fitted separately before 1995 and after 1995. The blue curve represents the number of bacteria genomes sampled each year (in logarithmic scale). Orange stars represent the dates at which the following aminoglycoside drugs were released for commercial use: streptomycin, neomycin, hygromycin, kanamycin, paromomycin, capreomycin, gentamicin, ribostamycin, dibekacin, tobramycin, amikacin, sisomicin, netilmicin, micronomicin, isepamicin, plazomicin. The red rectangle displays the period of time analyzed in panel: (**C**) Evolution of worldwide CHG frequencies among sampled AME genes between 1990 and 2019. 17 CHGs for which less than 400 sequences were sampled are grouped and displayed in gray.

The online version of this article includes the following figure supplement(s) for figure 4:

*Figure 4 continued on next page*

*Figure 4 continued*

**Figure supplement 1.** Evolution of continental CHG frequencies among sampled AME genes between 1990 and 2019.

**Figure supplement 2.** Distribution of sampling over time, space, ecology, and phylogeny.

APHa), but a negative significant effect for AACc and APHf Our proxy for potential bacteria influx due to migration has a negative significant effect on the probability to sample AME bacteria for AACd, AACf1, and APHf, and a positive effect for AACb and AACc. No significant effect of trade and migration were found for other CHGs. Despite the explanatory importance of ecology and the numerous interactions identified, for most CHGs, the probability of sampling AME bacteria does not significantly differ between most biomes. However, the probability of sampling AME bacteria in clinical samples

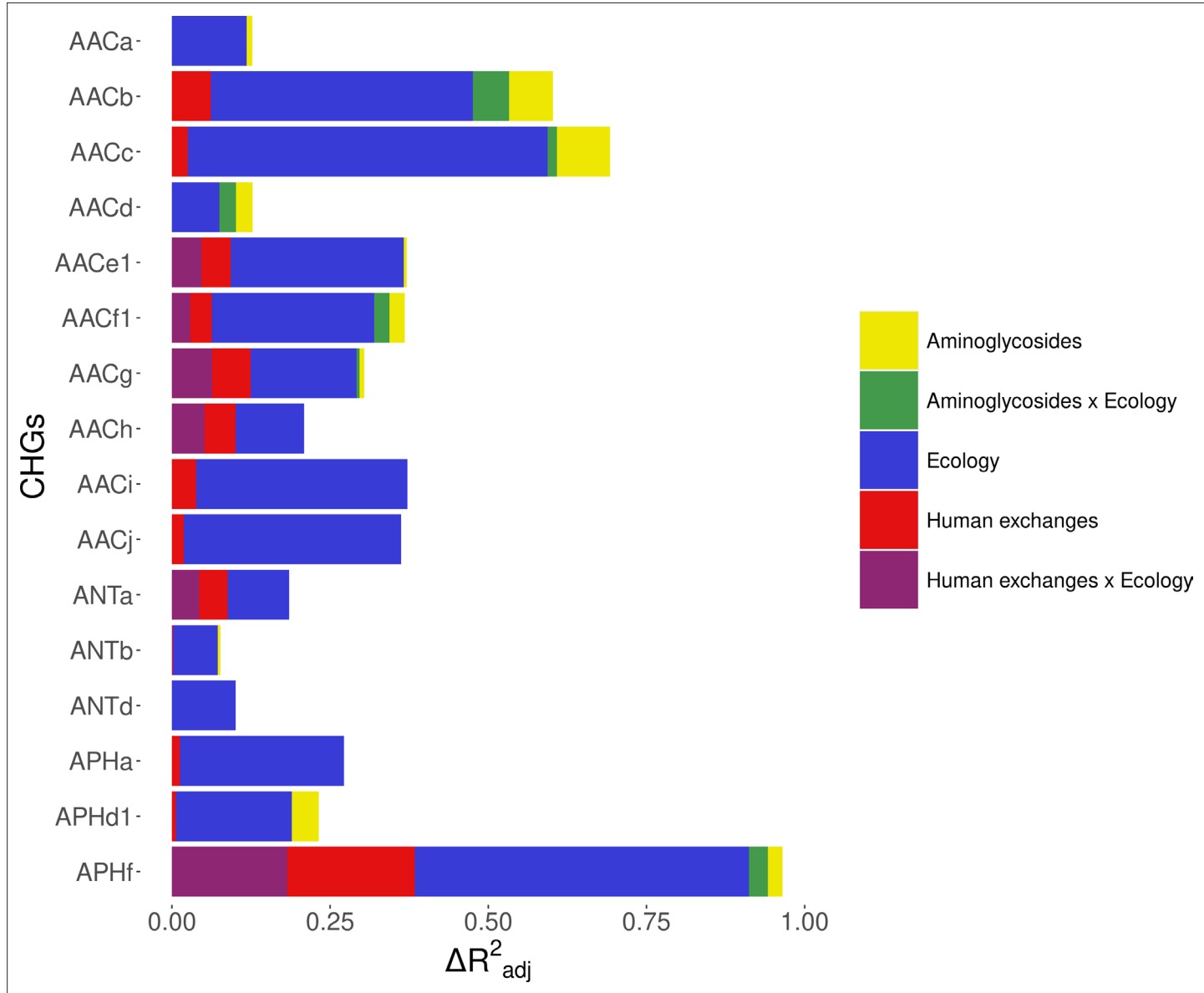

**Figure 5.** Relative importance of several factors to explain the distribution of aminoglycoside-resistant bacteria in Europe between 1997 and 2018. Logistic regressions with a spatial Matérn correlation structure were computed to explain the frequency of 16 CHGs. This figure represents the contribution of each variable in each selected model, as the fraction of adjusted McFadden's pseudo-$R^2$ explained by adding this variable.

The online version of this article includes the following figure supplement(s) for figure 5:

**Figure supplement 1.** Relative importance of several factors to explain the distribution of aminoglycoside-resistant bacteria in the world between 1997 and 2018.

is significantly higher for AACa, AACc, AACe1, and AACh (respectively significantly lower for AACb and ANTa) than in other biomes. In the same way, AACb, ANTa, and APHf are significantly less likely to be sampled in farms, while AACc is significantly more likely and AACb less likely to be sampled in soil. Regarding interaction effects, aminoglycoside consumption can have a positive significant effect on the probability to sample AME bacteria in clinical samples (for AACb, AACf1, and APHf), soil (for AACf1), and farms samples (for AACb). Human exchanges also have ecology-specific impacts for four CHGs only. For AACf1, trade has a negative effect on the probability to sample AME bacteria in clinical, farms, human, and soil samples, whereas migration has positive effects in clinical, farms, and human samples and a negative effect in freshwaterAME bacteria. For AACg, migration has a positive effect in clinical samples, whereas for ANTa, trade has a negative effect in soil samples. For APHf, migration has a negative effect in clinical, farms, and human samples, while trade has a positive effect in farms and a negative effect in clinical samples. To sum up, differences in aminoglycoside use are less likely to explain differences in aminoglycoside resistance prevalence, than sampling across different biomes.

We proceeded to the same analyses for worldwide data on the same timeframe, analyzing only the effects of ecology and human exchanges (see *Supplementary file 2*). Antibiotic consumption data could not be accessed at this spatial scale, but the same trend was found for the other factors: ecology is always the most important explanatory factor, and human exchanges are conserved as an explanatory factor for most CHGs (see *Figure 5—figure supplement 1*).

## What is the effect of biased sampling?

However, since our dataset consists of genomes sequenced for multiple research projects, it was necessary to test the effect of a sampling bias on these results: indeed, it can be easily assumed that sampling effort in different habitats will be focused on different bacterial taxa, for example those that have the greatest impact in these habitats. To account for this bias, the dataset was split into two subsets, each making up for approximately 50% of sampled genomes in each biome: one consisting of the 99 most sampled species (e.g. *Clostridioides difficile* in sludge and waste), and the other of the 805 least sampled species (e.g. *Lactobacillus kalixensis* in the clinical environment). This implies that a given species can be represented among either the most sampled or the least sampled species, depending on the biome in which it was sampled. A new model selection was computed on each of these datasets for the 7 CHGs for which 30 AME bacteria were available in both datasets (i.e. AACc, AACe1, AACf1, AACg, AACh, ANTa, and APHf). The results of this model selection are detailed in *Supplementary file 3*. The inputs of the models have a better explanatory power for the most sampled species dataset (on average $R^2_{adj}$ = 52.7%, ranging from 19.7% to 86.6%) than for the least sampled species dataset (on average $R^2_{adj}$ = 18.3%, ranging from 7.1% to 25.6%). Ecology is still the most important variable to explain variations in AME bacteria prevalence, being kept as an explanatory factor for all 7 CHGs in the case of highly sampled species, and 6 of them for the least sampled species. However, its importance varied greatly between the datasets: ecology increased $R^2_{adj}$ by 45.6% on average for most sampled species, but only by 14.8% for least sampled species. Human exchanges still accounted for variation in prevalence of 6 CHGs in the case of most sampled species (with an average increase of $R^2_{adj}$ of 3.4%), whereas for least sampled species, it had a higher importance (average increase of $R^2_{adj}$ of 4.5%) but for 4 CHGs only. Similarly, although aminoglycoside consumption contributed to the variance of 3 and 5 CHGs respectively for most and least sampled species, it had a higher importance for least sampled species (with an average increase of $R^2_{adj}$ of 6.3%) than for most sampled species (respectively 1.9%). Finally, interactions between ecology and human exchanges increased $R^2_{adj}$ by 3.2% for most sampled species (for 5 CHGs) and by 5.5% for least sampled species (for 3 CHGs), whereas interactions between ecology and aminoglycoside consumption increased $R^2_{adj}$ by 2.9% for most sampled species (for one CHG only) and by 3.3% for least sampled species (for 4 CHGs). Thus, although ecology is always the most important variable to explain the distribution of AME bacteria, especially compared to aminoglycoside consumption, its importance is overestimated by an unbalanced sampling of taxa across ecosystems.

## The effect of human exchanges

For the 13 CHGs sampled in Europe for which we kept human exchanges as explanatory variables in models of section 2.2.1, we computed new models based on the decomposition of these effects: we

replaced the two effects beneath human exchanges (i.e. international trade and immigration) in the model by only one effect: the proxy for either bacteria influx due to imports of one of the products considered, or bacteria influx due to immigration. We did not find any variable that was able to explain the distribution of all these CHGs, not even for a majority of them (see *Supplementary file 5*). At most, 2 categories of imports were able to impact the distribution of 5 CHGs with a positive effect (beverages, medicinal and pharmaceutical products), and 4 categories were able to explain the distribution of 4 CHGs with a positive effect (cereals, animal feedstuff, miscellaneous edible products, and sugar). However, the distribution of 4 CHGs correlated the most with one category of imports only: tobacco for AACc; oil seeds for AACf1; animal feedstuff for AACg; and human migration for APHf.

## The worldwide distribution of aminoglycoside resistance is mainly driven by ecology

Ecology being the most important explanatory factor for the distribution of AME genes, we investigated the distribution of the different CHGs across ecosystems. Here we approximated an ecosystem as the intersection of a biome and a geographical division (here using the IMAGE24 subdivision from the R *rworldmap* library, *South, 2011*). We described the aminoglycoside resistome composition profile of an ecosystem as the presence or absence of each CHG, in 5-year time frames between 1990 and 2019. For example, in clinical samples of Western Europe, ANTc is present over the 2015–2019 period and not before, while AACj is always present except during the 1995–1999 period. We measured differences in resistome composition as Jaccard index, and evaluated their correlation with

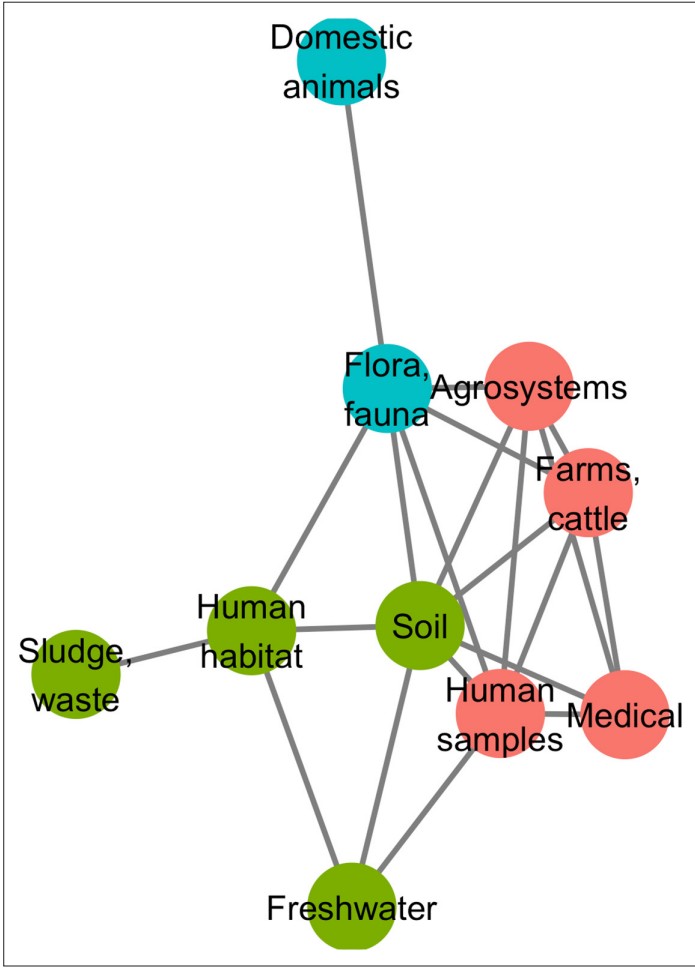

**Figure 6.** Minimum spanning network based on shared antibiotic resistance genes between biomes. Resistomes are represented here as vertices. Edges correspond to particularly high resemblance between two resistomes. Three modules in the network were established by the Louvain clustering algorithm: vertices are colored based on the module to which they belong.

ecological similarities (being 1 if two ecosystems belong to the same biome, 0 else) and geographical distances between ecosystems. On a global scale, there is no correlation between resistome composition and geography (Mantel test, 9999 permutations, Z=9.23.10⁹, p=0.585) but AME gene compositions depend on ecology (Mantel test, 9999 permutations, Z=174.9, p=1.10⁻⁴). Ecosystems are thus more similar in terms of AME gene content if they are from the same biome than if they are geographically close.

A network approach was adopted to get a deeper understanding of the ecology of the resistome composition. We converted the matrix of resistome composition similarities into a network where each vertex consists of a resistome and edges are weighted by Jaccard indices. A minimum spanning network was built that only retained similarity-based edges that could form maximum similarity paths between resistomes. The resulting network is displayed in *Figure 6*. Based on the Louvain clustering algorithm, we found three main ecological modules in this network, clustered by their composition in CHGs: (1) clinical, farms, human, and agricultural samples; (2) soil, human habitat, waste, sludge, and freshwater; (3) domestic animals and wild plants and animals. In this network of resistome compositions, wild plants and animals have the highest betweenness centrality (i.e. it is an intermediate in 14 shortest paths in the network), followed by the human habitat (8 shortest paths), soil (7 shortest paths), and human samples (3 shortest paths). All the other biomes have a null betweenness centrality. These four biomes also have the highest closeness centrality in the network. But degree centrality is actually highest for soil (i.e. it shares connections with 7 other biomes), followed by wild plants and animals and human samples (6 connections each), and by agrosystems and farms (5 connections each). Soil, wildlife, and human samples are thus the three biomes for which resistomes are more similar to each other over several time frames.

We then investigated whether resistome resemblance within each of the ecological modules defined above is correlated to phylogenetic similarity in bacteria community composition (defined as the whole set of bacteria sampled within a geographical and ecological unit, and measured with the phylogenetic Sørensen index). After correction for the difference in modules sample sizes, it was established that resistome resemblance is significantly, positively correlated with bacteria community composition similarity for both module 1 (clinical, farms, human, and agricultural samples, 95% CI on Pearson correlation coefficient, $\rho \in$[0.075, 0.335]) and module 2 (soil, human habitat, waste, sludge, and freshwater, 95% CI on Pearson correlation coefficient, $\rho \in$[0.042, 0.278]), but no significant correlation was found for module 3 (domestic animals and wild plant and animals, 95% CI on Pearson correlation coefficient, $\rho \in$[–0.042, 0.294]).

## Resistance gene accumulation is under selection for widening resistance spectra

Among 38,523 AME bacteria, 6246 (~16%) carried more than one AME gene, and the maximum number of AME genes in a genome was eight. We investigated whether and how this accumulation of AME genes within genomes lead to a broadening of the resistance spectrum. To do so, we used in separate analyses the classifications provided by two published datasets: either one empirically predicting the resistance phenotypes associated to several AR genes (*Feldgarden et al., 2019*), or another reviewing and compiling all the antibiotics to which most AMEs were reported to confer resistance (*Zárate et al., 2018*). Two classifications were used because *Feldgarden et al., 2019* probably underestimated each resistance spectrum (due to tests on a limited number of antibiotics), while *Zárate et al., 2018* probably overestimated them (due to an exhaustive review from heterogeneous sources). The classification by *Feldgarden et al., 2019* allowed to infer the resistance spectrum of 11,284 genomes from the 12,982 AME genes they carried, whereas the one by *Zárate et al., 2018* allowed to infer the resistance spectrum of 26,006 genomes with 31,910 AME genes. Using the resistance spectra inferred from *Feldgarden et al., 2019*, two classical functional dissimilarity indices (functional dispersion, FDis, and Rao's quadratic entropy, RaoQE) indicated that the functional diversity increases, that is resistance spectrum widens, with the number of resistance genes carried by a genome (FDis, linear regression, $R^2$=0.756, p<2.2.10⁻¹⁶ and RaoQE, linear regression, $R^2$=0.746, p<2.2.10⁻¹⁶). More importantly, these increases occur at a higher rate than expected under the hypothesis of a random assortment of AME genes among genomes, both for FDis (permutation test, 500 perms., p=1.2.10⁻³) and for RaoQE (permutation test, 500 perms., p=6.7.10⁻³). The same trends were also observed using the resistance spectra inferred from *Zárate et al., 2018*.

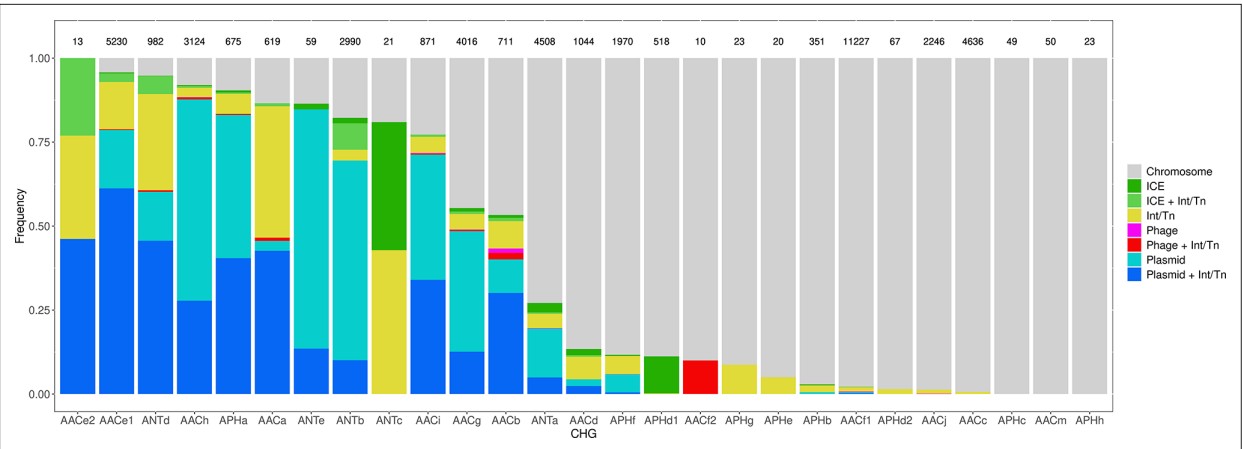

**Figure 7.** Distribution of the frequency of genes carried by MGEs per CHG. CHGs are organized by descending total frequencies of genes associated with MGEs. The number of genes identified for each cluster is indicated above each bar.

Among the 46,053 AME genes detected in the initial screen, 15,374 were detected to be associated with MGE conferring intergenomic mobility (plasmids, prophages, or ICEs), and 9640 were detected to be associated with a MGE conferring intragenomic mobility (integrons or transposable elements). A total of 7277 genes were detected to be associated with both at the same time. The proportion of genes associated with MGEs is actually very heterogeneous between CHGs, covering the full range from 0% to 100% (*Figure 7*). We thus investigated the role of MGEs on the accumulation of AME genes. FDis and RaoQE increase both with the number of MGE-associated AME genes and with the number of AME genes not associated with MGEs. Yet, the increase rate is significantly higher than expected under the hypothesis of random assortment for genes associated with MGEs conferring intergenomic mobility (permutation test, 500 perms., p=1.2.10⁻³ for FDis, p=1.2.10⁻³ for RaoQE), whatever the source used to infer resistance spectra. However, for AME genes associated with MGEs conferring intragenomic mobility or not associated to MGE, the trends differ depending on how the resistance spectra were inferred, and no conclusion can be drawn. The pattern of resistance spectrum widening with the accumulation of AME genes in a genome is thus stronger and more robust for genes associated with intergenomic mobility MGEs.

Many MGEs are also known to replicate the genes they carry in the same genome. We thus investigated which pairs of AME genes could be the result of intragenomic replication. The probability for an AME gene to undergo intragenomic replication is at least 10 times higher when associated with MGEs, under the hypothesis of random MGE-gene assortment (permutation test, 500 perms., p=1.2.10⁻³). This probability is similar when considering separately MGEs with intergenomic or intragenomic mobility.

## Discussion
### AME genes show a ubiquitous presence in sequenced bacteria genomes

In this study, we defined and screened 27 clusters of genes encoding resistance to aminoglycosides across more than 160,000 genomes spread across phylogeny, ecology, geography, and time. About one quarter of these bacteria were found to carry a gene known to provide resistance to aminoglycosides. This very high prevalence of aminoglycoside resistance is likely biased by the much higher availability of genomic data for human, clinical, and agriculture isolates. Yet, the lowest prevalence found in a biome was actually 9%, which is consistent with previous studies on genomic (*Pal et al., 2015*; *Zhang et al., 2020*) and metagenomic (*Zhang et al., 2020*) datasets sampled from multiple biomes. Around 40% of the AR genes found were potentially mobile, that is associated with plasmids, integrons, prophages, ICEs or transposons. A previous study reported <20% of potentially mobile AR genes: 29.4% for AR genes conferring multidrug resistance, 15.5% for β-lactam resistance, and

10.5% for aminoglycoside resistance (*Zhang et al., 2020*), but this study only focused on plasmids and integrons.

AME bacteria are widely spread over space, time, and ecology. We were able to detect AME genes in all the biomes considered, and all over the world. The prevalence of AME bacteria ranged from 64% in Turkey to 0% in Arctic and Antarctic regions. AME bacteria were found as early as 1905, that is nearly 40 years before the first aminoglycoside was isolated and identified as a potential antibacterial therapeutic agent. This is not a surprise as aminoglycosides are naturally produced by bacteria from the *Streptomyces* and *Micromonospora* genera (*Durand et al., 2019*) and the selection pressure for the evolution of resistance genes existed long before the clinical use of aminoglycosides, as shown for other antibiotic resistance genes (*D'Costa et al., 2011*).

AME bacteria prevalence strongly increased between the 1940s and the 1980s, and has plateaued around 30% since the 1990s (*Figure 4B*). The phase of increasing prevalence can easily be explained by the discovery and marketing for clinical use of most aminoglycosides. Their massive use must have strongly selected for the emergence and spread of AME genes. However, AG consumption has stabilized and/or decreased (at least since 1997 in Europe), so the pre-existing selection pressures have remained stable since the 1990s. Besides, most CHGs coexist over long time periods (*Figure 4C*), but they are unevenly distributed across space (*Figure 2B*). This apparent coexistence might therefore result from the combination of different local dynamics (*Figure 4—figure supplement 1*). Interestingly, two very distant regions can display the same time trends: for example, the replacement of AACf1 by AACc and AACe1 in both Europe and Southern Asia and Oceania since the 1990s, or a surge in the frequency of ANTb in both Europe and North America during the 2000s. This suggests that local distributions of AME genes are driven by local parameters, but that local resistomes are also connected at the global scale. Finally, the ecological and phylogenetic distribution of AME genes varies to a great extent, from ubiquitous CHGs to some being restricted to a few biomes and species. Ubiquitous CHGs were first sampled earlier than the others, and the extent of ecological and phylogenetic spread for each CHG is positively correlated to the time since its first sampling occurrence. It is however impossible to determine whether a broader spread made it more likely to be detected earlier or an early emergence gave a CHG more time to spread across several biomes and phyla.

## Antibiotic resistance prevalence is shaped not only by antibiotic use but also by ecology and human exchanges

Reduction of antibiotic consumption and fighting antibiotic pollution have historically been the most common public health recommendation to control or reduce the spread of antibiotic resistance (see e.g. *Mölstad et al., 2008*; *Sabuncu et al., 2009*). The predominance of this recommendation is driven by the idea, in evolutionary epidemiology, that selection is the major determinant in the emergence, accumulation, and propagation of resistance (*Blanquart et al., 2017*; *Spicknall et al., 2013*). This idea is supported by correlative studies linking antibiotic consumption levels and resistance prevalence, mainly in environments where antibiotic consumption is high such as hospitals or farms (e.g. *Goossens et al., 2005*).

In this study, one of the goals was to ask whether the impact of aminoglycoside use on aminoglycoside resistance prevalence could be inferred on a very broad geographical and temporal scale and also across biomes. The local concentrations of aminoglycosides are likely to strongly vary between biomes, as antibiotics are mainly used in hospitals, outpatients, farms and agrosystems and antibiotics are found as pollutant in other biomes. Data on the amount of antibiotics prescribed in human and veterinary clinics are generally available but antibiotic pollution data are globally sparse (although local databases have been set up in recent years, see *Umweltbundesamt, 2016*) and were not available on a sufficiently broad geographical and temporal scale to be integrated in our analysis. Thus, here we considered that aminoglycoside concentration is sufficiently correlated between biomes (as suggested for other antibiotics by e.g. *Li et al., 2016*), since a large part of the antibiotics can pollute other biomes (*Danner et al., 2019*; *Gothwal and Shashidhar, 2015*; *Kümmerer and Henninger, 2003*; *Tello et al., 2012*), so that variation in aminoglycoside prescriptions have the same effects throughout the biomes. However, one could alternatively consider that aminoglycoside consumption data are only reliable in the biomes where they are directly used, because aminoglycoside concentrations differ between hospitals and farms on the one hand, and soil, freshwater, etc. on the other hand. Hence we performed another model selection in this case to control for this potential effect,

by only including biomes where antibiotics are expected to exert a strong selection pressure that is clinical, human, and farms samples (for a total of 15 CHGs, see *Supplementary file 4*). The selected models accounted on average for a reduction of $R^2_{adj}$=28.5% of deviance. On this subset, even though a lower number of biomes was considered, the contribution of aminoglycoside consumption to the prevalence of AME genes is small in comparison to differences between biomes: ecology was kept as an explanatory factor for 15 CHGs and had an average contribution of 18.8% to $R^2_{adj}$, whereas aminoglycoside consumption and its interaction with ecology explained the prevalence of respectively 9 and 4 CHGs (with average contributions of respectively 2.1% and 1.0% to $R^2_{adj}$).

Thus, independently of the biomes considered, our results show that between 1997 and 2018 in Europe, aminoglycoside consumption is not the main factor explaining AME genes prevalence patterns. For all CHGs studied, aminoglycoside consumption was only a minor explanatory factor for the distribution of AR bacteria, with few positive effects and mainly non-directional effects on the probability to sample AR bacteria. There are even in this dataset examples of low ARB prevalence occurring in countries with high antibiotics consumption: for example over the period, the average prevalence of bacteria carrying AACf1 was 1.5% in France and 18.1% in Austria, whereas these two countries consumed on average respectively 0.22 and 0.02 g per inhabitant/human-equivalent of aminoglycosides each year. Our results clearly contrast with the previously established correlations between antibiotic consumption and antibiotic resistance prevalence as well as the obvious selective advantage conferred by resistance genes across large range of antibiotic concentrations (*Gullberg et al., 2011*). One reason could be that the period studied (1997–2018) is likely to be a post-emergence period during which aminoglycoside consumption was on average stable in Europe. The relative weight of selection as evolutionary force driving the resistance gene frequency variation is likely to be lower than in the previous period. Additionally, several mechanisms can allow resistant strains to thrive even under decreasing antibiotic selection pressure. (i) Selective pressures on AR are multiple and man-made antimicrobial agents are just one of them. Many AMEs might have evolved as adaptations to antibiotics produced in natural communities, or as exaptations of metabolic enzymes selected for other functions (e.g. AACs from enzymes involved in the acetylation of ribosomal proteins, *Rather, 1998*). (ii) AME genes can be maintained in populations by hitchhiking on other genes (e.g. resistance to heavy metals, biocides) carried on the same MGE (*Zhai et al., 2016*). (iii) Carrying an AR gene is costly only if the gene is expressed, but AR genes can be silenced in nonselective conditions and be reactivated in selective conditions only (*Kime et al., 2019*). The results of our integrated study strongly suggest that reducing aminoglycoside consumption is a necessary but not a sufficient measure to limit the propagation of antibiotic resistances and that complementary measures based on the reduction of other selection pressures (such as heavy metal pollution, *Seiler and Berendonk, 2012*) maintaining AR gene frequency should be implemented.

## Trade and migration matter more than antibiotic use to explain antibiotic resistance prevalences

In our dataset, human exchanges explain a significant part of the variation in the prevalence of AME bacteria for a higher number of CHGs than antibiotics consumption does. Over the 1997–2018 period, we observed this trend within Europe, and human exchanges also have an important effect worldwide (*Figure 5—figure supplement 1* and *Supplementary file 2*), although at this larger scale it was impossible to compare their effects to the effect of antibiotic consumption. Numerous examples of direct impact of human activities, outside of antibiotic use, on the emergence, retention, and propagation of AR have been documented: AR bacteria can be carried over continents by plant (*Zurfluh et al., 2015*) and animal products (*Eltai et al., 2020*; *Le Hello et al., 2011*), exchanged through international trade, as well as immigrants (*Nellums et al., 2018*) and travelers in general (e.g. *Lepelletier et al., 2011*). The AR genes carried by imported strains may then spread to local strains and species by horizontal transfer, thus enriching the local resistome or increasing the local frequency of resistant genes. In the case of AME genes, the decomposition of human exchanges in different good categories revealed that no category, either specific (e.g. animal feedstuff imports) or general (e.g. animal products imports), contributes to explain the distribution of all CHGs. This could be due to different CHGs having different geographical origins, to different CHGs being carried by bacteria with different ecological niches, or to contingency in the different transmission pathways. Yet, these factors were identified by correlation only, so whether these imports are transmission pathways for these CHGs

cannot be assessed without direct sampling of such products. Once this assessment will be done, the similarity in resistome composition between the exporting and importing countries could technically become a criterion for the choice of the origin of some imported products.

The importance of migration of AR genes, through trade and human travels, as a factor explaining the variation in AR gene frequency, revealed by our analysis suggests that reducing the import of AR bacteria would be an efficient way of limiting AME gene propagation. Procedures of AR gene monitoring in traded goods have been set up but are probably insufficient. For example, as reported for the European Union (*Schrijver et al., 2018*), the modalities for detection and characterization of AR genes in meat are not harmonized between countries, and only focus on clinically relevant species, thereby ignoring the risk of horizontal transmission to and from other unmonitored strains.

## Ecology matters most to explain antibiotic resistance prevalence

Not only did we find a wide diversity of AME genes outside of hospitals and farms, where antibiotics are the most consumed (see *Figure 3*), but differences between biomes explained most of the variance when the frequency of AR bacteria was modeled over time, space, and ecology. Across ecosystems (defined as the intersection of a biome and a geographical unit), we found that the resemblance between resistomes depended more on the biomes in which they were sampled, than on their geographical location. This is consistent with previous studies: functional metagenomic selections show drastic differences in the resistance profiles between soil and human gut microbiota isolates (*Gibson et al., 2014*), and <10% of AR genes sampled in wastewater treatment plants can also be found in other environments (*Munck et al., 2015*). Yet, consistent with *Forsberg et al., 2012* who found AR genes shared with 100% identity between soil bacteria and human pathogens, many AR genes were still shared between biomes, including between natural and anthropized biomes.

Additionally, the study of certain biomes seems crucial to understand the spread of AR. Indeed, in the network of resistomes, 'fauna and flora' and 'soil' have the highest measures of centrality, which suggests that these two biomes are involved in most events of AME gene transfer across ecology (either as an origin or as a destination). This could be explained by both antibiotics and AR genes transferred from humans and anthropized habitat to natural ecosystems. Natural ecosystems have actually been suggested to constitute reservoirs for AR genes originating from hospitals (see e.g. *Baquero et al., 2009*; *Tripathi and Cytryn, 2017*). Also, most natural aminoglycosides are produced by two bacteria genera, *Streptomyces* and *Micromonospora* (*Durand et al., 2019*), that live mostly in soil and decaying vegetation. It is therefore possible that many of the CHGs first evolved in soil and flora bacteria as defense mechanism against the chemical warfare of these two genera and remained at high frequency because of continuous selection pressure. Along the same lines, we found that, in our dataset, even though 19 times more genomes were sampled in hospitals than in soil, only 2.5 times more CHGs had their first documented occurrence in clinical samples than in soil. If not all CHGs emerged in soil and wildlife, these two biomes appear at least as a hub for the circulation of AR genes.

Our biome network approach identified three major modules of resistome: (1) hospitals, humans, farms, and agrosystems; (2) soil, human habitat, waste, and freshwater; and (3) domestic and wild animals and plants. Previous metagenomic studies using abundance-based metrics found that resistome composition is correlated to bacteria community composition: either globally (*Pehrsson et al., 2016*), or in specific biomes: soil (*Forsberg et al., 2014*), wastewater treatment plants (*Ju et al., 2019*), human feces (*Pehrsson et al., 2016*). In agreement with these studies, while using presence-absence-based indices, we found substantial correlation between resistome similarity and phylogenetic beta-diversity in the first and the second modules. This correlation indicates that resistome similarity is at least partially due to species composition similarity, and so to the exchange of AR bacteria between biomes. Consequently, our results also indicate that the propagation of AME genes could result from the dissemination of AR bacteria: between ecosystems. The propagation could occur by abiotic dissemination (*Allen et al., 2010*), fecal transmission (*Karkman et al., 2019*), transmission by wastewater (*Vaz-Moreira et al., 2014*), or transmission through the food chain (*Founou et al., 2016*). Yet, the correlation between resistome and species composition was not found in the third module, even though we controlled for the lower sampling in this module. This surprising result could actually be due to a higher environmental fragmentation in these biomes (see e.g. *Lu et al., 2014* for the gut microbiota of wild flying squirrels), which could result in biased sampling. Moreover, previous results showed that higher spatial structures increase the likelihood of HGT events to low-abundance

strains (*Cairns et al., 2018*), that might not be sampled and therefore decrease the likelihood to find a correlation between resistome composition and community composition. The absence of correlation between resistome and species composition leads to formulate the hypothesis that horizontal transfer might play a more important role in AR gene propagation in this module than in the other two. This hypothesis could be tested using HGT detection methods such as the one developed in *Corel et al., 2018*.

## Combination of AME genes in a genome is shaped by diversifying selection and MGEs movements

In our dataset, 40% of AME genes are carried by MGEs. MGE carriage is known to be a strong determinant of the capacity to propagate within and between genome and has been identified as key element for classifying a given antibiotic resistance in the highest risk category (*Martínez et al., 2015*). The association with MGE frequently takes the form of an embedded structure: for example half of transposable elements carrying AR genes were also located on plasmids (*Figure 7*). The embedded structures combine the intra- and inter-genome propagation potential. At a finer scale, the fraction of genes associated to MGE as well as the identity of the associated MGE strongly differ between CHGs, suggesting that these clusters have contrasted propagation probabilities and pathways.

The MGE carriage of resistance genes also means that they can be acquired from different sources and combined within a genome. A second interesting finding is that the combination of AME genes within genomes is not random: the resistance spectrum widens at a higher rate with the number of AME genes than expected by chance, which suggests that the combination of AME genes in a genome is under selection for functional diversification. This broadening of the resistance spectrum is mainly driven by MGEs with inter-genomic mobility (i.e. plasmids, ICEs, and prophages).

Finally, we also found that MGEs increase the likelihood for a genome to carry several copies of the same AR gene. It has to be noted here that the copy number accessible from whole genome sequence data is an underestimation of the actual number of copies of the gene for all genomes in which the AR gene is carried by a plasmid, because plasmids are usually present in more than one copy. The presence of several copies of a resistance gene, either because of its association with an intra-genomic MGE or its plasmid carriage not only increases the expression level of AR genes (*Depardieu et al., 2007*; *Sandegren and Andersson, 2009*), but also allows bacteria to evolve new antibiotic resistance functions on duplicated sequences (*San Millan and MacLean, 2017*; *Sandegren and Andersson, 2009*), thus participating in the functional diversification.

## Limitations

The approach taken here allows to exploit a large amount of publicly available data to gain a broad scale vision of AME gene circulation and propagation. However, it suffers some drawbacks, mainly linked to the fact that it is based on available data and does not result from a dedicated sampling. On the one hand, our screening is likely to underestimate the frequency of resistance carrying genomes because some AME families probably remain unknown and aminoglycoside resistance can be conferred by other resistance mechanisms such as target change, hydrolysis, etc. (*Blair et al., 2015*). On the other hand, resistance gene carrying genomes do not necessarily produce resistant bacteria. It is indeed nearly impossible to determine with certainty whether each of the genes identified is expressed and codes for a functional protein and thus actually confers aminoglycoside resistance. So far, the genotype-phenotype relationship is poorly understood for AR genes in general (*Hughes and Andersson, 2017*), and particularly for AMEs: a single amino acid change is likely to change the enzyme's target, and even to remove its resistance function (*Feldgarden et al., 2019*; *Zárate et al., 2018*). However, our screening might still be considered as a decent approximation when working with a high spatio-temporal scale of genomic data.

Other limitations come from the fact that our dataset consists of genome sequences in multiple research projects. Because of this, sampling was biased towards industrialized countries and towards phyla and biomes with clinical interest (see *Figure 4—figure supplement 2* for a full overview). Thus, the genomes available are not representative of the species composition of the different biomes and some of the biomes or geographical location are over-represented. The frequencies of resistant bacteria established in this study are useful for comparisons between biomes, geographical location and time periods but cannot be taken as absolute estimates. In particular, the diversity of

CHGs represented across phyla can be assumed to be explained by the differences in the number of genomes sampled for each phylum, which is itself linked to the presence of species of medical interest in some phyla. Along the same line, the absence of detected resistance in certain places in our data set is likely due to a lack of published genomic data from these places: for example, we did not detect any AME gene in genomes of bacteria from the Antarctic, when metagenomic studies have shown that AG resistance have evolved in polar communities (*Perron et al., 2015*). However, the repetition of the analyses on several subsets of the dataset to take sampling bias into account showed that our main conclusions still hold on each of the subsets.

Besides, environmental data were insufficiently standardized. Regarding sampling locations, many of them were unknown in NCBI Biosamples metadata; and others could only be determined to the precision of the country, sometimes by looking at unrelated columns. We thus chose to use the country scale as spatial grain, in order to consider as many of them as possible, but at the cost of geographical precision. Moreover, because sampling can be scarce in many geographical areas, countries outside Europe were grouped in larger entities. This choice of spatial scales, constrained by the metadata available, might have prevented to uncover dynamics that occur at finer spatial scales. In the same way, we had to rely on unstandardized data regarding the categorization of the biomes in which bacteria were sampled. Our categorization of biomes was an attempt to reproduce current categorization of metagenomes, but unlike metagenomic datasets, the ecosystems in which bacteria genomes are sampled are usually poorly described: for example the distinction between human samples and clinical samples is often very subtle. Some samples may therefore have been assigned to the wrong biome. And since sampling is greatly biased toward clinical, human, and farms samples, we chose to merge certain classical ecological categories in order to treat the widest diversity of samples. This categorization thus partly differs from other studies.

## Conclusion and perspectives

The present study provides a broad picture of the spatial, temporal and ecological distributions of AME genes as well as their association with MGEs and reveals contrasted patterns for the different gene families. It additionally establishes that the recent temporal variations of AME bacteria in Europe are explained first by ecology, second by human exchanges and last by antibiotic consumption. This means that selection by man-made antibiotics is not the only evolutionary force explaining the frequency of aminoglycoside resistance and its variation, such that interventional strategies based on prudent uses of aminoglycosides for humans, animals, and plants are likely to be a necessary but insufficient way to control and limit the spread of aminoglycoside resistance. The importance of ecology and human exchanges in shaping the patterns of AME gene prevalence is adding to the growing body of evidences that AR depends not only on clinical therapeutic guidelines, but also on the high interconnectivity of ecosystems, both locally and globally (*Hernando-Amado et al., 2019*). Thus, although continuing AR monitoring in clinical and farms samples is crucial, current sampling methods hugely bias genomic datasets and insufficient standardization of data limits their exploitation. Understanding the big picture on AR will require a stronger sampling effort in natural ecosystems and as we have shown that resistomes tend to cluster by ecology rather than by geography, emphasis should be put on monitoring the resistome of all biomes with equal intensity. Although the conclusions of this study cannot be extended to AR genes other than AMEs, the methods implemented could easily be applied to other AR gene families (especially modifying enzymes). As a complement, this could help us clarifying the overview of the forces that shape resistance prevalences across large temporal and spatial scales.

Finally, this study highlights that AME genes are frequently associated with MGE but also shows that this level of association strongly varies between gene families. It additionally reveals the role of MGE in the generation of within-genome duplications and even more importantly in functional diversification and resistance spectrum broadening. MGEs are known to be vehicles of HGT and are likely to participate in the spread of AME genes but the strong correlation between resistome composition and species composition established here within biome groups suggests that AME genes are also spreading by AME bacteria between biomes. The relative contributions of HGT and bacteria migration to AR propagation as well as the factors that shape and orient them should be investigated on large, integrative datasets in order to understand the antibiotic resistance traffic rules.

## Materials and methods

### Detection of aminoglycoside modifying enzyme genes

The first aim of this study was to document the geographic, historical and ecological extension of gene families that encode AMEs. This was done by screening for these genes across genomes of the clade of Eubacteria. Current methods to detect AR genes in genomic datasets can be, for example, based on HMM profiles drawn from biochemical nomenclatures (e.g. AMRFinder, *Feldgarden et al., 2019*). However, the biochemical characteristics of AMEs do not reflect their evolutionary history and many profiles may thus gather both homologous and paralogous sequences. Such mixed and evolutionary heterogeneous profiles do not allow to track the spread of a gene family as they can gather sequences with different evolutionary origins despite their similar biochemical functions. New HMM profiles based on clusters of homologous sequences only were thus built. While it may detect a wider diversity of resistance genes, it also increased the need for additional filtering steps.

### Definition of resistance profiles

204 sequences of aminoglycoside modifying enzymes were gathered from examples reviewed in *Ramirez and Tolmasky, 2010* and *Garneau-Tsodikova and Labby, 2016*. From the sequences contained in these two references, two categories have been excluded for further analyses: sequences for which (i) the protein could not be found from the provided identifier solely, and (ii) two enzymatic functions were assigned (e.g. AAC(6')-APH(2")). All the other sequences (*Source data 4*) were grouped in 29 clusters of homologous genes (CHGs) using SiLiX (v. 1.2.9, *Miele et al., 2011*) with the following parameter values: at least 70% overlap and 35% identity to accept BLAST hits for building families. For each CHG, sequences were aligned with Clustal Omega (v. 1.2.4, *Sievers and Higgins, 2014*), and alignments were submitted to HMMER (v. 3.3, *Eddy, 2011*) to define profiles. Sequences used to define CHGs, their biochemical classification, and their assignment to CHGs, are listed in *Source data 4*: AACs were split into 14 CHGs, ANTs into 6 CHGs, and APHs into 9 CHGs, and no CHG gathered proteins with distinct enzymatic activities.

### Bacterial genomes

The genomes included in this study are all the Eubacteria genomes available in the NCBI Refseq database on August 18, 2019 and that could be assigned at the species taxonomic level, to the taxonomy *bac120* (release 89, *Parks et al., 2018*) with the pipeline GTDB-tk (v. 1.0.2, *Chaumeil et al., 2020*). This pipeline extracts sequences of 120 conserved genes to assign unannotated genomes to the taxonomy, based on the sequence of these 120 genes. 160,987 genomes were used in this study.

### Screening resistance profiles into genomes

The genomes were screened for the presence of the 29 CHG profiles defined using the *hmmsearch* command of HMMER (v. 3.3). Around 1,200,000 hits were found in the 160,987 genomes screened. Results were kept with cutoff values of $10^{-3}$ for e-value and 0.95 for accuracy, and if the length of the predicted proteins was at least 80% of the one of the corresponding profile. When the same protein corresponded to two (or more) overlapping profiles, the result with the lowest e-values was kept. The functions of the proteins screened by HMMER were predicted using InterProScan (v. 5.40, *Jones et al., 2014*) and proteins with a predicted function incompatible with aminoglycoside modification were filtered out of the dataset based on a keyword list search. On the one hand, sequences that may have an enzymatic function similar to the ones of AMEs were screened with the following keywords: 'acetyltransferase', 'adenylyltransferase', 'adenyltransferase', 'phosphotransferase', 'phosphoryltransferase', and 'nucleotidyltransferase'. On the other hand, sequences whose functions might be involved in aminoglycoside resistance were screened with the following keywords: 'aminoglycoside', 'aminoside', 'mycin', 'micin', and 'amikacin'.

After screening and functional filtering, some CHG profiles did not match any output homologs (i.e. ANTf, APHi, AACk, AACl, and AACn) and they were thus excluded: this left only 24 CHGs. The proteins conserved after this functional filtering step were reclustered with SiLiX (with a minimum of 40% identity over 80% overlap) in order to confirm homology. As some sub-clustered CHGs also contained a low number of hits (2 or 3), only clusters that contained at least 10 sequences were kept (<100 sequences excluded). Three CHGs had to be subdivided and the resulting subclusters will be

referred to as their initial CHG name, plus another digit (e.g. AACf1): this additional clustering added 3 CHGs to the 24 remaining ones. This resulted in 46,053 protein sequences predicted to code for AMEs, belonging to 27 CHGs, and spread across 38,523 genomes.

## The genomic context of aminoglycoside modifying enzyme genes

A search for mobile genetic elements in the neighborhood of predicted AME genes was performed. The software programs and methods used are detailed thereafter. All programs mentioned below were applied to complete genomes for completely assembled genomes, or to each contig containing AME genes from partially assembled genomes. The aim was to identify MGEs allowing intra-genomic mobility (transposable elements and integrons) and those allowing inter-genomic mobility (plasmids, prophages, and ICEs). Importantly, about one third of the identified AME genes were located less than 1 kb of the extremities of the corresponding contigs. The association of these genes with MGEs might have been greatly under-detected, depending on the detection method: whether it is based on each contig in its entirety (see below for plasmids, for which detection should not be reduced), or whether it is based on the upstream and downstream neighborhood of each focal gene (see below for composite transposon, for which detection should be nearly impossible).

### Identification of genetic elements allowing inter-genomic mobility

The PlasForest pipeline (v. 1.0.0, *Pradier et al., 2021*) was used to identify plasmids. Full contigs containing previously identified AME genes were submitted to this random forest classifier-based software. When a contig was predicted to be of plasmid origin, all the AME genes it carried were considered as carried by a plasmid. Prophages were then identified with the pipeline PhiSpy (v. 4.0.0, *Akhter et al., 2012*). AME genes were considered to be associated with prophages when located within a predicted prophage, or within 1 kb upward or downward of it. At last, integrative and conjugative elements (ICEs, sometimes referred to as 'conjugative transposons') were identified with the pipeline ICEFinder (*Liu et al., 2019*). This pipeline can only process annotated genomes, so AME-gene-carrying genomes were beforehand annotated with the software Prokka (v. 1.14.5, *Seemann, 2014*). AR genes were considered to be associated with ICEs when located within a predicted ICE, or within 1 kb upward or downward of it.

### Identification of genetic elements allowing intra-genomic mobility

There are two types of bacterial transposable elements that are likely to carry accessory genes: composite and noncomposite transposons. However, there is currently no standardized method to detect transposable elements in prokaryote genomes (see *Goerner-Potvin and Bourque, 2018*, with the exception of Red that is not specific to transposable elements, *Girgis, 2015*). Thus, in order to detect composite transposons, ISfinder database of insertion sequences (*Siguier et al., 2006*) was downloaded (2020/05/05) and used as a BLASTn subject database (v. 2.11.0, *Camacho et al., 2009*). Among the hits found in AME-gene-carrying contigs, only those larger than minimum IS size (depending on the family, from 600 bp for IS200/IS605 to 3000 bp for Tn3) were kept. An AME gene was considered as potentially carried by a composite transposon if it was surrounded by at least 2 IS of the same family within 20 kb from each other. In order to detect noncomposite transposons, 345,657 transposase protein sequences were extracted from NCBI (list of accession numbers in Supplementary information List. S1) and used as a BLASTp subject database. Contigs carrying AME genes were translated on 6 frames, then submitted to BLASTp against this database, where only hits larger than 50 amino acids were kept (in order to detect only hits with substantial coverage of transposases). In order to distinguish IS transposases (involved in composite transposon transposition) from other transposases, AME genes in the neighborhood of IS (previously identified as associated to composite transposons) were excluded from this noncomposite transposon screen. Remaining AME genes were considered as potentially associated with noncomposite transposons if they were located within 20 kb from a transposase.

At last, integrons were identified with the pipeline IntegronFinder (v. 1.5.1, *Cury et al., 2016*). AME genes were considered to be associated with integrons when located within a predicted integron, or within 1 kb upward or downward of it.

## Environmental contexts of AR gene-carrying bacteria

### Metadata

Through the NCBI BioSamples database, metadata were collected for 90,751 samples (about 55% of the genomes screened for AME genes).

A classification was attempted for all the samples into the 11 ecological units (biomes) that were defined: clinical environments; freshwater; sludge and waste; wild fauna and flora; domestic animals; farms and farm animals; agrosystems; sea water; human habitat; human samples; soil. To do so, keywords were searched, in a precise order to avoid multiple allocation of the same sample. Among the BioSamples metadata columns, a number of columns were isolated to search for information enabling to characterize a sampling biome for each genome. On the one hand, were selected the column whose names contained one of the following keywords: 'biome', 'habitat', 'env', 'host'. On the other hand, other columns were selected if they contained the keyword 'source' or 'isolate', but none of the following keywords: 'date', 'year', 'by', 'id', 'preservation', 'number', 'energy', 'carbon', 'method', 'comment', 'note', 'resource', 'history', 'annot', or 'lab'. Samples were then classified into biomes based on the contents of these columns (see *Supplementary file 6*).

Some form of sample location was available in the metadata of 75,581 samples: either in the form of geographical coordinates (13,053 samples), location names (exact location, region, or country, 52,898 samples), or institution (university, laboratory, or hospital, 7970 samples). As most locations were not more precise than the country scale, each sampling location was assigned to a country by reverse search in Google Geocode API. Sea water samples could not be assigned to any country, and were thus not included in analysis of spatial variation in AME genes prevalence. Due to the large number of geographical areas to consider, countries were gathered into larger regions (IMAGE24 classification, R library *rworldmap*, *South, 2011*). Samples originating from European Union member countries kept their country-level assignment for analyses focusing on Europe only. Sampling dates recovered from the metadata were considered to the time level of year.

In total, geographical information, ecological information, and sampling dates could be recovered for 45,574 genomes only. Because sea water samples could not be assigned to any country, they were excluded from the dataset.

### Antibiotic consumption

Aminoglycoside consumption data in the European Union were collected from the ESAC-NET (for human consumption, *European Centre for Disease Prevention and Control, 2021*) and the ESVAC (for animal consumption, *European Medicines Agency, 2021*) databases for the period 1997–2018. Antibiotic consumption data are not freely available for countries outside the European Union. As an aim of this study was to test the impact of antibiotic consumption in general, both sources of antibiotic consumption were summed into a single variable, measured in grams of aminoglycosides consumed each year by each country. ESVAC consumption data were measured in metric tons, while ECDC consumption data were reported in Defined Daily Doses (DDDs) for 1000 habitants. ECDC consumption data were thus converted to the same unit using 1 g as a baseline for aminoglycoside DDD, which is the current WHO standard for amikacin (*WHO Collaborating Centre for Drug Statistics Methodology, 2020*), and each country's population, recovered from the *World Bank, 2021a*. In order to standardize consumption data between countries (and time periods) of heterogeneous populations, these measures were divided by each country's human and animal population. Farm animal populations (with the exception of equines for which too little data were available) over the period were recovered from the Eurostat website (*Eurostat, 2022*). Missing animal population data (e.g. breaks in the time series due to changes in measurement method) were predicted by interpolating the actual data between two consecutive data points (linear regression: Population ~Animal type x Country x Year, $R^2=0.989$). However, since animals with very different weights can receive very different doses of antibiotics, a common unit (coined as 'human'equivalent") was established to standardize the total population of farm animals. As an approximation, individual weight was considered to be 3 kg for poultry, 80 kg for goats and sheep, 200 kg for pigs, and 700 kg for cattle: for example it would take approximately 27 chickens to make up the weight equivalent of one human. Then, missing human and animal consumption data on the period 1997-2018 were predicted by interpolating the actual data between two consecutive available data points (linear regression: Antibiotic consumption in grams ~Country x Year, $R^2=0.959$ for ESVAC data, $R^2=0.704$ for ECDC data). Finally, these data were summed

to a single variable of AG consumption per country and per year, and divided by the total human and human-equivalent population.

## Human exchanges

The bilateral trade matrix, including imports between 1997 and 2018, was collected from the United Nations Conference on Trade and Development (*United Nations Conference for Trade and Development, 2021*). This matrix measured imports in 2018 US dollars. From this matrix, imports of energy, information, and services were excluded, to only keep 27 import categories (categories are listed in the rows of *Supplementary file 5*). The bilateral immigration matrix was collected from the *World Bank, 2021b*. Missing immigration data were predicted by interpolating the actual data (linear regression: Number of migrants ~Country of origin x Country of destination x Year, $R^2$=0.883).

Potential AME gene influx from one country to another was calculated as the sum of imports (respectively immigrants) from the region of origin, multiplied by the frequency of AME genes sampled in the region of origin. Total AME gene influx in a country due to trade (respectively immigration) was computed by summing influxes from all possible origins.

## Inference of resistance spectrum

The aim was to assign a resistance spectrum to each of the AME gene we identified. AR phenotypes in particular can be either determined by empirically testing the resistance phenotype associated to a given genotype as has been done in the AMRFinder database (*Feldgarden et al., 2019*), or inferred from the systematic compilation of all the antibiotics to which a biochemical family is known to resist (*Zárate et al., 2018*). Here both datasets were used to infer a minimum and maximum resistance spectrum of the AME genes detected in Eubacteria genomes. An AME biochemical class was attributed to each AME gene sequence identified thanks to the AMRFinder database. A resistance spectrum was then attributed to each AME gene sequence using the established correspondence between the biochemical classes and the AR phenotypes, either empirically predicted (*Feldgarden et al., 2019*), or reviewed (*Zárate et al., 2018*). Even if an assignment of each AME gene sequence to an AMRFinder profile could be done, an assignment of each of them to a resistance spectrum was not possible, either because no precise resistance was indicated in *Feldgarden et al., 2019*, or because the biochemical classes corresponding to AMRFinder profiles were too broad compared to the ones from *Zárate et al., 2018* and thus did not allow correct assignment. An AR phenotype could be assigned to 12,982 AME genes out of 11,284 genomes using empirically tested data, and to 31,910 AME genes out of 26,006 genomes using reviewed data.

## Statistical analysis

### Phylogenetic diversity

To evaluate the phylogenetic range across which each CHG is distributed, two measures summing up the phylogenetic distribution were computed using the function *pd* from the R library *picante*: (i) species richness, that is the number of species in which a CHG was found; and (ii) Faith's distance (*Faith, 1992*), that is the sum of the lengths of all the branches on the bac120 tree that span bacteria carrying a CHG.

### Factors influencing the distribution of aminoglycoside resistance

For each CHG, the aim was to explain the local frequencies of AG-resistant bacteria, that is the likelihood to sample a genome carrying at least one gene from a given CHG in a given ecosystem at a given date, during the 1997–2018 period. The function *fitme* of R library spaMM (*Rousset and Ferdy, 2014*) was used to compute logistic regressions. Due to large variation in sampling across time, countries and biomes, in the models, the local frequencies were weighed by the number of sampled genomes they were computed with. The random structure of each null model was selected to minimize conditional AIC, among these hypotheses: no structure; a first-order time-autoregressive process; a Matérn correlation structure based on longitude and latitude of each country's centroid; both a time-autoregressive process and a spatial Matérn correlation structure. Several hypotheses were also considered for the explanatory variables. First, AME gene influx due to trade and AME gene influx due to immigration were highly correlated with each other. In the subsequent statistical analysis, these two variables were thus not considered separately: either both of them were considered to explain the

distribution of antibiotic resistance, or none of them were. These two variables were thus considered under the denomination of human exchanges. Then, the following variables were included as fixed effects: ecology (as a categorical variable), AG consumption, human exchanges, and their interactions with ecology (as scaled numerical variables). Including the null model, thirteen different combinations of fixed effects were thus simultaneously considered to explain the distribution of a CHG. For each CHG, the model with the lowest conditional AIC was selected. As antibiotic consumption data were only available for Europe, this model selection was only performed for European data. Due to the temporal variability of sampling in Europe, dates were considered every two years. Three additional model selections were performed. (i) The first one, without considering the antibiotic consumption variable, for worldwide data (with countries being gathered into larger geographical units, see section 4.3.1) with dates being considered every year. (ii) The second one on European data, but split into two datasets based on each species' sampling across biomes. (iii) The third one on European data, but excluding data sampled outside of clinics, farms, and humans. In each of these cases, models were performed only on CHGs that were carried by at least 30 genomes in the dataset considered.

To describe the amount of variance explained by the models and the improvement of the complete models from the null models, adjusted McFadden pseudo-$R^2$ were computed:

$$R^2_{Adj} = 1 - \frac{log(L) - K}{log(L_0)}$$

where $L_0$ is the likelihood of the null model, $L$ is the likelihood of the considered model, and $K$ is the number of explanatory variables in the model. To measure the unique contribution of each variable, the difference $\Delta R^2_{Adj}$ was computed, that is the difference of adjusted McFadden pseudo-$R^2$ value measured in the same model without the focal variable.

For 13 CHGs, model selection kept human exchanges as one of the explanatory variables. For these cases, the effect of the 28 components of the 'human exchanges' variable was tested. All these components being highly correlated with each other, 'human exchanges' was replaced by only one component at a time. Among the 28 resulting models, only those for which the slope estimates for human exchanges were positive were kept. Finally, models with AIC $<AIC_{min}10$ were selected (**Burnham and Anderson, 2002**).

## Distribution of resistomes across geography and ecology

In our dataset, an ecosystem was defined as the intersection of a geographical region and a biome. An aminoglycoside resistome (hereafter simplified as 'resistome') was defined as the set of AME genes that could be found in an ecosystem. Several CHGs appeared during the time frame considered (1990–2018). Thus, in order for resistomes not to depend on time, CHG presences and absences over several time periods were separately considered (from 1990–1994, 1995–1999, etc.). For example, AACa presence/absence between 1990 and 1994 and its presence/absence between 1995 and 1999 are considered as different variables for resistome composition comparisons, as if they were two independent CHGs. Similarities between resistomes of different ecosystems were computed using Jaccard index, because it is one of the least sensitive to sampling error (see **Schroeder and Jenkins, 2018**). This index takes a value of 1 when two ecosystems carry exactly the same CHGs, a value of 0 when two ecosystems carry no common CHG.

To test the dependence of resistomes on geography and ecology, distances were considered between ecosystems (i) either as geographical distances between the centroids of considered regions, or (ii) as binary distances between considered biomes (measured as 1 if two ecosystems belong to the same biome, 0 otherwise). Mantel tests were performed to measure a possible correlation between Jaccard index on one side and the geographical or the biome distance on the other side. Correlation with ecological binary distance was found but not with geographical distance, so ecosystems were then grouped into biomes for further analyses.

To test the association strength between biomes, a network based on the previous matrix of Jaccard indices was created. Each biome was represented as a vertex, and each edge was weighed by the resemblance between two resistomes (measured by Jaccard index). This complete network was pruned into a minimum spanning network (**Bandelt et al., 1999**), in order to remove the edges that were the least likely to correspond to AR gene exchange between two resistomes. Modules were established on this network by using the Louvain method (**Blondel et al., 2008**). Finally, the relationship between AME gene composition and bacteria community composition was studied at the

ecosystem level (each ecosystem being defined as the intersection of a biome and a geographic unit). To do so, the correlation between the resistome Jaccard index and the similarity in bacteria community composition, represented by a phylogenetic Sørensen index (*Bryant et al., 2008*), was computed.

### Functional diversity

*Functional dispersion* (FDis) and *Rao's quadratic entropy* (RaoQE) were computed to characterize functional dissimilarity, in terms of resistance spectrum conferred by different AME genes contained in the same genome. For this, the R library FD was used (*Laliberté et al., 2014*). To measure how functional dissimilarity changes with AME gene number per genome, a linear regression of either the square root of FDis or the square root of RaoQ and the logarithm of the number of AME genes carried by a genome was computed. Permutation tests were performed in order to compare the observed pattern with expectations under the null hypothesis of random assortment of AME genes among genomes. The gene-genome matrix was permuted 500 times using the function *permatswap* from the R library *vegan*, and FDis and RaoQ were calculated on each genome for each iteration. For each permutation, linear regressions were computed, which yielded a distribution of slopes, simulated under the null hypothesis of random assortment of AME genes among genomes. Observed slopes were then compared to this distribution, to assess whether AME gene accumulation is compatible with this hypothesis. This analysis was performed for whole genomes, and then distinguishing between AME genes associated or not associated with MGEs and then further distinguishing between MGEs with intragenomic and intergenomic mobility.

### Gene duplications

For each genome that contains at least two AME genes belonging to the same CHG, the DNA sequences of these genes were recovered, as well as 1 kb upstream and downstream, from the NCBI Nucleotide database. Pairwise alignment of these sequences was performed, in order to measure sequence identity between pairs of AME genes. Because many gene sequences are identical or nearly identical, sequence identity was also computed between pairs of sequences composed of genes and their close genomic contexts. As a conservative threshold, a group of sequences (e.g. a pair, a triplet, or a quadruplet of sequences) was considered to be the result of gene duplication if AME genes were at least reciprocally 90% identical and their genomic contexts were at least reciprocally 80% identical. Yet, to test the sensitivity of the subsequent analysis to these arbitrary thresholds, other thresholds were used for the identity between AME genes (80%, 85%, 90%, and 95%) and for the identity between genomic contexts (50%, 70%, 80% and 90%). Results did not substantially vary with these thresholds (*Supplementary file 7*). To test if the association of an AME gene sequence with (at least) one MGE makes it more likely to duplicate, the proportions of duplicated groups of AME genes were computed both among genes associated with MGEs and among genes that were not. Permutations of the AME genes-MGEs matrix were performed to compute replicates of these proportions under the hypothesis of random physical association between MGEs and AME genes. The observed odds ratio was then compared to the distribution obtained by permutation.

## Acknowledgements

We thank Christelle Leung and Tazzio Tissot for their helpful comments that helped improve the manuscript, and Alexandre Courtiol and Martijn Callens for stimulating scientific and technical discussions. This work was supported by the ERC HGTCODONUSE (ERC-2015-CoG-682819) to SB.

## Additional information

### Funding

| Funder | Grant reference number | Author |
| --- | --- | --- |
| European Research Council | ERC-2015-CoG-682819 | Stéphanie Bedhomme |

| Funder | Grant reference number | Author |
|--------|------------------------|--------|

The funders had no role in study design, data collection and interpretation, or the decision to submit the work for publication.

## Author contributions

Léa Pradier, Conceptualization, Data curation, Formal analysis, Investigation, Visualization, Methodology, Writing – original draft; Stéphanie Bedhomme, Conceptualization, Supervision, Funding acquisition, Validation, Project administration, Writing - review and editing

## Author ORCIDs

Léa Pradier (ID) http://orcid.org/0000-0002-1545-3585
Stéphanie Bedhomme (ID) http://orcid.org/0000-0001-8075-0968

## Decision letter and Author response

Decision letter https://doi.org/10.7554/eLife.77015.sa1
Author response https://doi.org/10.7554/eLife.77015.sa2

# Additional files

### Supplementary files

• Supplementary file 1. Description of the models selected to explain the distribution of aminoglycoside resistant bacteria in Europe between 1997 and 2018.

• Supplementary file 2. Description of the models selected to explain the distribution of aminoglycoside resistant bacteria in the world between 1997 and 2018.

• Supplementary file 3. Description of the models selected to explain the distribution of aminoglycoside resistant bacteria in Europe between 1997 and 2018 under low or high sampling.

• Supplementary file 4. Description of the models selected to explain the distribution of aminoglycoside resistant bacteria in Europe between 1997 and 2018, in biomes where antibiotics are prescribed.

• Supplementary file 5. Human exchanges variables influencing the prevalence of aminoglycoside resistant bacteria.

• Supplementary file 6. Classification of biomes according to the keywords of the NCBI BioSamples database.

• Supplementary file 7. Impact of the association of AME genes with mobile genetic elements with their duplication risk.

• Transparent reporting form

• MDAR checklist

• Source data 1. List of genomes that were screened for aminoglycoside resistance.

• Source data 2. List of AME genes that were found by screening.

• Source data 3. Distribution of screened AME genes over time, geography, and ecology.

• Source data 4. AME sequences used to define clusters of homologous genes.

### Data availability

All data generated or analysed during this study are available as Source Data Files 1, 2, and 3. Accession numbers of sequences analyzed during this study are provided in Source Data Files 1, 2, and 4. The softwares used to generate these data are all publicly available and the parameters used are detailed in the Methods.

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
