## [Editor Report]

This work presents important findings on the role of ecological niches in shaping the distribution of antibiotic resistance genes in the environment and human populations. The evidence supporting the conclusion is compelling, presenting a rigorous analysis of a large dataset with many possible confounding variables. The work should be of broad interest to microbiologists, public health workers, and policymakers.

---

## [Decision Letter]

**Decision letter after peer review:**

Thank you for submitting your article "Ecology, more than antibiotics consumption, is the major predictor for the global distribution of aminoglycoside-modifying enzymes" for consideration by *eLife*. Your article has been reviewed by 2 peer reviewers, and the evaluation has been overseen by a Reviewing Editor and Meredith Schuman as the Senior Editor. The reviewers have opted to remain anonymous.

The reviewers have discussed their reviews with one another, and the Reviewing Editor has drafted this to help you prepare a revised submission. In your revision, please directly address the essential revisions. You may draw on the individual reviews to help you address these, but a point-by-point response to each reviewer comment is not needed.

Essential Revisions:

1) The authors should justify their focus on aminoglycosides as opposed to other classes of antibiotics.

2) The reviewers expressed concerns regarding the fact that aminoglycosides were rarely used in humans over the period considered in the study. The authors should address how this is possibly impacting their conclusions regarding antibiotic consumption in humans as a factor driving the spread of aminoglycoside resistance.

3) The authors should provide additional statistical analysis on the possible impact(s) of sampling biases on their conclusions. For example, the authors could use subsampling techniques to evaluate to what extent their results and conclusions are due to biases in their dataset.

*Reviewer #1 (Recommendations for the authors):*

A more detailed analysis of the biases in the collected dataset, and how these biases influence the main conclusions of the study, is needed to strengthen the study's conclusions. Here are some suggestions:

A multi-panel supplementary figure showing the breakdown in the metadata e.g. Numbers of genomes from different phyla, habitats, year, country, etc. Aminoglycoside-resistant bacteria from different phyla, habitats, years, country, etc. Individual CHGs by habitat, year, country, etc. This would help direct the reader to the biases evident in the data.

Inclusion of a sampling bias statistic as an explanatory variable within the models.

Robustness of the model to subsampling of the data, weighted sums, considerations of only over-sampled and under-sampled taxa, etc.

The language could be tightened and sentences shortened in a few places to enhance interpretation. For example, in the abstract, it is not readily clear to me what this sentence means: "soil, wildlife, and human samples appear to be central to understand the exchanges of AMEGs between different ecological contexts"

I did not find any justification for the focus on aminoglycosides antibiotics.

Final sentence: "The relative contributions of HGT and ACB migration to AR propagation as well as the factors that shape and orient them should be investigated on datasets as the one generated in this study in order to understand the antibiotic resistance traffic rules and potentially interfere with them to reduce and slow down their movements."

I find the acronyms did not help my readership of the paper. Here are some: ARB, ARG, ARM, AR, AG, AMEs, AMEGs, ACBs, CHG. They are all so similar. I suggest just using AR for antimicrobial resistance, CHG for clusters of homologous genes, and getting rid of all the others to enhance readability.

*Reviewer #2 (Recommendations for the authors):*

Smaller recommendations and suggestions for improvement are also listed below:

1. There are many areas where the text can be consolidated, especially in the discussion.

2. It would be good to put the CHG families into some type of organizational context. As is, the numbers are hard to follow and it's very hard to take meaning away from, for instance, lines 198-221. Can the figure axes organize CHGs by enzymatic type, date of discovery, or some other useful grouping?

3. There were too many acronyms, which made the text hard to read.

4. Figure 2 is prominent in the manuscript and its discussion seems to imply a relationship between resistome composition and geography. Yet, on lines 251-252, this is apparently not the case. I'd consider focusing the figures on the key, most salient points.

5. Why was the seawater biome omitted from figure 3? Why does CHG26 show no presence in any biome?

6. It would be useful to succinctly summarize the key meaning of highly technical paragraphs, in a single concluding sentence. For example, on line 302.

7. The terms 'One Health' and 'Global Health' perspectives are presented as if their meaning is obvious when this will not be the case for many readers. The subsequent sentences do a fine enough job encapsulating their key concepts. I think the inclusion of these terms serves to confuse and I suggest removing the relevant sentence.

8. Line 656 refers to 29 CHGs, but the text writ large discusses only 27. Where are the missing two?

9. Could the metadata biome terms be included as a supplemental table? In sentence form, as is written in the current methods section, it is very hard to parse.

---

## [Author Response]

Thank you for submitting your article "Ecology, more than antibiotics consumption, is the major predictor for the global distribution of aminoglycoside-modifying enzymes" for consideration by eLife. Your article has been reviewed by 2 peer reviewers, and the evaluation has been overseen by a Reviewing Editor and Meredith Schuman as the Senior Editor. The reviewers have opted to remain anonymous.The reviewers have discussed their reviews with one another, and the Reviewing Editor has drafted this to help you prepare a revised submission. In your revision, please directly address the essential revisions. You may draw on the individual reviews to help you address these, but a point-by-point response to each reviewer comment is not needed.Essential Revisions:1) The authors should justify their focus on aminoglycosides as opposed to other classes of antibiotics.2) The reviewers expressed concerns regarding the fact that aminoglycosides were rarely used in humans over the period considered in the study. The authors should address how this is possibly impacting their conclusions regarding antibiotic consumption in humans as a factor driving the spread of aminoglycoside resistance.3) The authors should provide additional statistical analysis on the possible impact(s) of sampling biases on their conclusions. For example, the authors could use subsampling techniques to evaluate to what extent their results and conclusions are due to biases in their dataset.

1. We have added a paragraph in the introduction to justify our focus on aminoglycosides as opposed to other classes of antibiotics in the introduction (see lines 88-102). On the one hand, aminoglycosides still are widely used as a topic treatment of otitis in humans, as a second-intent drug for patients infected by multi-resistant bacteria, and they still represent a large share of antimicrobials used for farm animals. Thus, aminoglycoside resistance is still a threat to human health, especially when facing a rise of resistance to first-intent treatments, and a threat to food production. Moreover, since most aminoglycosides originate from soil-dwelling bacteria, AMEs should spread between at least three ecological contexts (hospitals, farms, and soil). They potentially represent a good model for resistance to antibiotics that also originated from the environment (e.g. β-lactams).

2. Following suggestions made by Reviewer #2, we have performed complementary analyses of the dataset to properly test the impact of aminoglycoside consumption on the distribution of aminoglycoside resistance genes. First, data on farm animal populations in Europe were collected, in order to normalize aminoglycoside consumption by capita rather than by land area (see lines 993-1011). Second, we had tried to palliate the lack of antibiotic pollution data by assuming that antibiotic concentration remain unchanged as they diffuse from one biome to the other, but Reviewer #2 showed us that this assumption had very little support in the literature. Thus, we created a subset of the European dataset, only including the biomes in which aminoglycoside consumption data should be reliable (clinical, human, and farms samples, see lines 307-334 in the Results section). We argue that restricting the analysis to these biomes should have allowed us to draw reliable estimates of the effect of aminoglycoside consumption on the spread of aminoglycoside resistance. We thus performed an additional model selection on this subset. This new analysis still supports our initial conclusion that variations in aminoglycoside consumption are very weakly associated with variations in AME-gene-carrying bacteria prevalences.

3. The binomial regressions we had performed on the European dataset (as well as on the worldwide dataset) already included a correction for low sampling: this type of models allows the precision of estimates to increase with sample size (the proportion confidence interval being correlated with sqrt(1/n)). However, as argued by Reviewer #1, there might have been sampling biases depending on ecology (e.g. Cyanobacteria being more sampled in water, *Enterobacteriaceae* in clinical environments, etc.), which could have affected our conclusions. We have thus performed complementary analyses of the European dataset to test this possibility. We splitted the European dataset into two subset: one containing only the most sampled species (relative to each biome), and the other containing the least sampled species (see lines 274-305 in the Results section). By performing two distinct model selections on these two datasets, we found that sample biases did have the effect of overestimating the contribution of ecology to explaining the spread of AMEs. However, our main conclusion still holds true: ecology still is the variable explaining most variation in the prevalence of AME-gene-carrying bacteria, especially compared to the effect of aminoglycoside consumption.

Additionally, as suggested by Reviewer #1, we added Figure 4—figure supplement 2 and Source Data File 3 so readers may have a better idea of sampling biases in our dataset.

4. For better clarity, we simplified the list of acronyms. Moreover, we renamed the list of clusters of AMEs, so that they are grouped by their broad biochemical function (N-acetyltransferase, nucleotidyltransferase, or phosphotransferase).

5. We rephrased several sentences in the discussion that could have suggested that our results would be applicable to antibiotic resistance in general and/or that antibiotic stewardship efforts could be unnecessary.